# Multi-omics and machine learning reveal context-specific gene regulatory activities of PML::RARA in acute promyelocytic leukemia

William Villiers[1], Audrey Kelly[2], Xiaohan He[1], James Kaufman-Cook[1], Abdurrahman Elbasir[3], Halima Bensmail[4], Paul Lavender [2], Richard Dillon [1,5], Borbála Mifsud [6,7] ✉ & Cameron S. Osborne [1] ✉

The PML::RARA fusion protein is the hallmark driver of Acute Promyelocytic Leukemia (APL) and disrupts retinoic acid signaling, leading to wide-scale gene expression changes and uncontrolled proliferation of myeloid precursor cells. While known to be recruited to binding sites across the genome, its impact on gene regulation and expression is under-explored. Using integrated multi-omics datasets, we characterize the influence of PML::RARA binding on gene expression and regulation in an inducible PML::RARA cell line model and APL patient ex vivo samples. We find that genes whose regulatory elements recruit PML::RARA are not uniformly transcriptionally repressed, as commonly suggested, but also may be upregulated or remain unchanged. We develop a computational machine learning implementation called Regulatory Element Behavior Extraction Learning to deconvolute the complex, local transcription factor binding site environment at PML::RARA bound positions to reveal distinct signatures that modulate how PML::RARA directs the transcriptional response.

Rearrangement at t(15;17)(q24;q21) between the *Promyelocytic Leukemia* (*PML*) and *Retinoic Acid Receptor Alpha* (*RARA*) genes results in the generation of a *PML::RARA* fusion gene, whose expression is the primary cause of acute promyelocytic leukemia (APL)[1,2]. Under normal physiologic conditions, wild-type RARA protein acts as a transcription factor/nuclear receptor controlling myeloid cell differentiation through the granulocytic lineage[3]. The binding of retinoic acid to RARA results in dimerization and subsequent activation of genome-wide retinoic acid-responsive elements (RARE), directing expression programs required for cell differentiation[4]. The PML::RARA fusion protein directly disrupts RARA signaling, leading to a differentiation block and proliferative drive[5].

Most focus on PML::RARA function centers on its dominant-negative role over canonical RARA action. PML::RARA retains the DNA-binding properties of the RARA transcription factor and the dimerization/oligomerization domain of the PML moiety, increasing DNA-binding capacity and co-factor recruitment potential[6]. PML::RARA can bind directly to gene promoters, where it can repress transcription through recruitment of co-repressors such as SMRT (silencing mediator of retinoid and thyroid hormone receptor), N-CoR (nuclear receptor corepressor), and RXR (retinoid X receptor), in addition to histone deacetylases/methyltransferases to silence gene transcription[6–9]. While RARE sites are suggested to be the prominent targets of PML::RARA, the oligomerization potential of PML::RARA extends the DNA-binding repertoire and allows complex DNA-binding

[1]Department of Medical and Molecular Genetics, King's College London, London, UK. [2]School of Immunology & Microbial Sciences, MRC and Asthma UK Centre in Allergic Mechanisms of Asthma, King's College London, London, UK. [3]ICT Division, College of Science and Engineering, Hamad Bin Khalifa University, Doha, Qatar. [4]Qatar Computing Research Institute, Hamad Bin Khalifa University, Doha, Qatar. [5]Department of Haematology, Guy's and St. Thomas' NHS Foundation Trust, London, UK. [6]College of Health and Life Sciences, Hamad Bin Khalifa University, Education City, Doha, Qatar. [7]William Harvey Research Institute, Queen Mary University London, London, UK. ✉e-mail: bmifsud@hbku.edu.qa; cameron.osborne9@gmail.com

configurations[10]. The PML: PML oligomerization ability also provides the potential for interactions with other transcription factors such as PU.1, SP1, and GATA2, adding additional dimensions of DNA-binding through recruitment of other DNA-binding proteins[11,12]. A full catalog of PML::RARA binding partners and their transcriptional effects is incomplete. Intriguingly and in contrast to its known repressive role, PML::RARA has been reported to directly activate *MYB* gene expression, suggesting divergent consequences of PML::RARA recruitment[13]. Indeed, more recently this non-canonical activity has been suggested to be more widespread[14]. Yet it is unclear how and why the consequence of PML::RARA recruitment is modulated.

Multiple groups have employed the inducible APL model cell line, U937-PR9, to better understand the early transformative events that occur following PML::RARA expression. In one study, PML::RARA binding positions were mapped by ChIP-seq, which highlighted that over 80% of PML::RARA binding positions were positioned outside the proximal promoter[15]. This may imply that PML::RARA is also recruited to distal regulatory elements, such as enhancers. The gene targets and contributions to APL pathogenesis of these non-promoter PML::RARA binding elements remain underexplored. Others also examined PML::RARA recruitment in the same experimental model, using a ChIP-chip strategy[12]. Significantly, there was very little agreement between the reproducible PML::RARA binding positions within these two studies, which suggests that further PML::RARA binding positions remain to be mapped.

In this study, we interrogate the impact of PML::RARA-mediated transformation by measuring the transcriptional response by RNA-seq, the genome-wide distribution of PML::RARA binding by Cut&Run, chromatin occupancy by ATAC-seq, and long-range interactions by promoter capture Hi-C. Through integration of these datasets and the distillation of machine-learned patterns using Regulatory Element Behavior Extraction Learning (REBEL), we identify complex pathways and transcription factor binding environments that support differing transcriptional outcomes in response to PML::RARA recruitment and are recapitulated for at least some crucial myeloid proliferation and differentiation genes in APL patient samples.

## Results

### PML::RARA induction instigates expression changes for thousands of genes

In this study, we investigated early transcription and gene regulatory events that occur in response to PML::RARA expression, through the integration of multi-omics measurements in U937-PR9 cells, a monocytic cell line that harbors a zinc-inducible PML::RARA fusion transgene[16] (Fig. 1a). The cell line was induced with zinc sulfate and incubated for five hours before being assayed for transcription, fusion protein binding, long-range promoter interaction, and chromatin occupancy. These conditions led to a PML::RARA protein increase that was approximately five-fold greater than levels detected in NB4 cells, a line with a constitutive PML::RARA rearrangement, and 1.4 to 2.5-fold higher than levels detected in three primary APL patients (Supplementary Fig. 1a).

We first examined the gene expression response to zinc induction. In agreement with PML::RARA protein expression quantitation, there was a strong induction of PML::RARA fusion transcripts (Supplementary Fig. 1b). Comparing uninduced and zinc-induced, highly reproducible replicate libraries, we identified 2314 differentially expressed genes (DEG), distributed roughly equally between up and downregulated genes (Fig. 1b and Supplementary Fig. 1c–f). These expression patterns were similar yet distinct from published datasets from the parental U937 cell line that lacks the PML::RARA transgene, and the NB4 cell line that contains a constitutively active t(15;17) PML::RARA rearrangement[17,18] (Supplementary Fig. 1c). Amongst the most highly differentially expressed genes in our datasets were those that had roles regulating zinc homeostasis, likely in response to the induction

treatment. Downregulated genes showed strong gene ontology enrichment for immune cell activation genes driving cell differentiation (Fig. 1c). These genes included *SPI1*, a master regulator of myeloid differentiation, and key myeloid differentiation driver genes *CEBPA*, *CEBPB*, *CEBPE*, *SP1*, and *ID2*. Upregulated genes were enriched for cancer pathways, cytokine production, and cell activation, and included the key *CEBP* antagonist, *BACH2*, as well as cell cycle progression genes, *CCNA1*, *CCNE2*, *CDKN2B*, and *CDKN1A*. Examination of all differentially expressed genes demonstrated a robust enrichment for proliferative signaling pathways such as NFKB, MAPK, and FOXO signaling, and pathways prominent in acute myeloid leukemia and other cancers (Fig. 1d). Together, these gene expression changes are consistent with patterns that are detected in APL.

### Transcriptional responses vary for genes whose promoters are bound by PML::RARA

We next examined patterns of PML::RARA fusion protein binding in response to zinc induction. Similar experiments have been carried out previously by ChIP-chip and ChIP-seq in two separate studies that identified PML::RARA binding sites by overlapping PML and RARA binding signals[12,15]. Notably, there was little overlap of PML::RARA binding sites between these studies (Supplementary Fig. 2a). We applied the Cut&Run method to identify PML::RARA binding sites due to its high signal-to-noise ratio and ability to accommodate antibodies not necessarily suited for ChIP[19]. In addition to an anti-RARA antibody, we also used one specific for the PML::RARA fusion protein. Both antibodies detected a strong, reproducible induction of binding upon zinc activation (Fig. 2a and Supplementary Fig. 2b–f), however, only the anti-RARA antibody detected reproducible peaks in the uninduced cells. We detected 1576 and 8712 peaks in the uninduced and induced samples, respectively, with a 45% overlap (Supplementary Fig. 2g). In contrast, by using the anti-PML::RARA antibody, we identified 99 and 15,412 peaks in the uninduced and induced samples, respectively (Supplementary Fig. 2h). The peaks detected in the uninduced cells were considerably weaker than those in the induced cells and may represent a thresholding artifact (Supplementary Fig. 2i). PML::RARA peaks in the induced sample overlapped with 86% of the induced RARA peaks, but markedly less so (31%) in the uninduced samples (Supplementary Fig. 2j, k). Using the anti-PML::RARA antibody, we detected 91% of the peaks detected by Martens et al., 64% of peaks detected by Wang et al. and an additional 10,902 peaks (Supplementary Fig. 2a). We also profiled PML::RARA binding in NB4 cells by Cut&Tag[20], identifying 20,072 PML::RARA binding sites (Supplementary Fig. 2l). Sixty-seven percent of the PML::RARA binding sites identified in induced U937-PR9 cells were detected in NB4 cells. The strong overlap indicates that higher PML::RARA expression in U937-PR9 cells does not lead to off-target recruitment. Collectively, these results demonstrate high specificity for the fusion protein by the anti-PML::RARA antibody, confirm the vast majority of previously identified PML::RARA binding sites, and identify thousands of additional sites.

We assessed the positions of all PML::RARA binding sites relative to genes. Nearly half of the sites (47%, 6727 genes) were located at gene promoters, with the remaining sites split between intronic (27%) and intergenic regions (24%) (Fig. 2b). Binding sites associated with promoters were typically stronger and wider than non-promoter peaks (Supplementary Fig. 2m, n).

Given its widely reported role as a transcriptional repressor, we anticipated that PML::RARA binding at promoters would be frequently associated with a drop in expression levels in the induced cells. However, upon the integration of our RNA-seq and Cut&Run datasets, we were struck by unexpected observations; firstly, 80% of genes whose promoters bound PML::RARA did not significantly change their expression, which suggests that PML::RARA binding has no consequence on the transcriptional output of most genes (Fig. 2c, d). This observation was not due to the stringent thresholding of DEGs, since

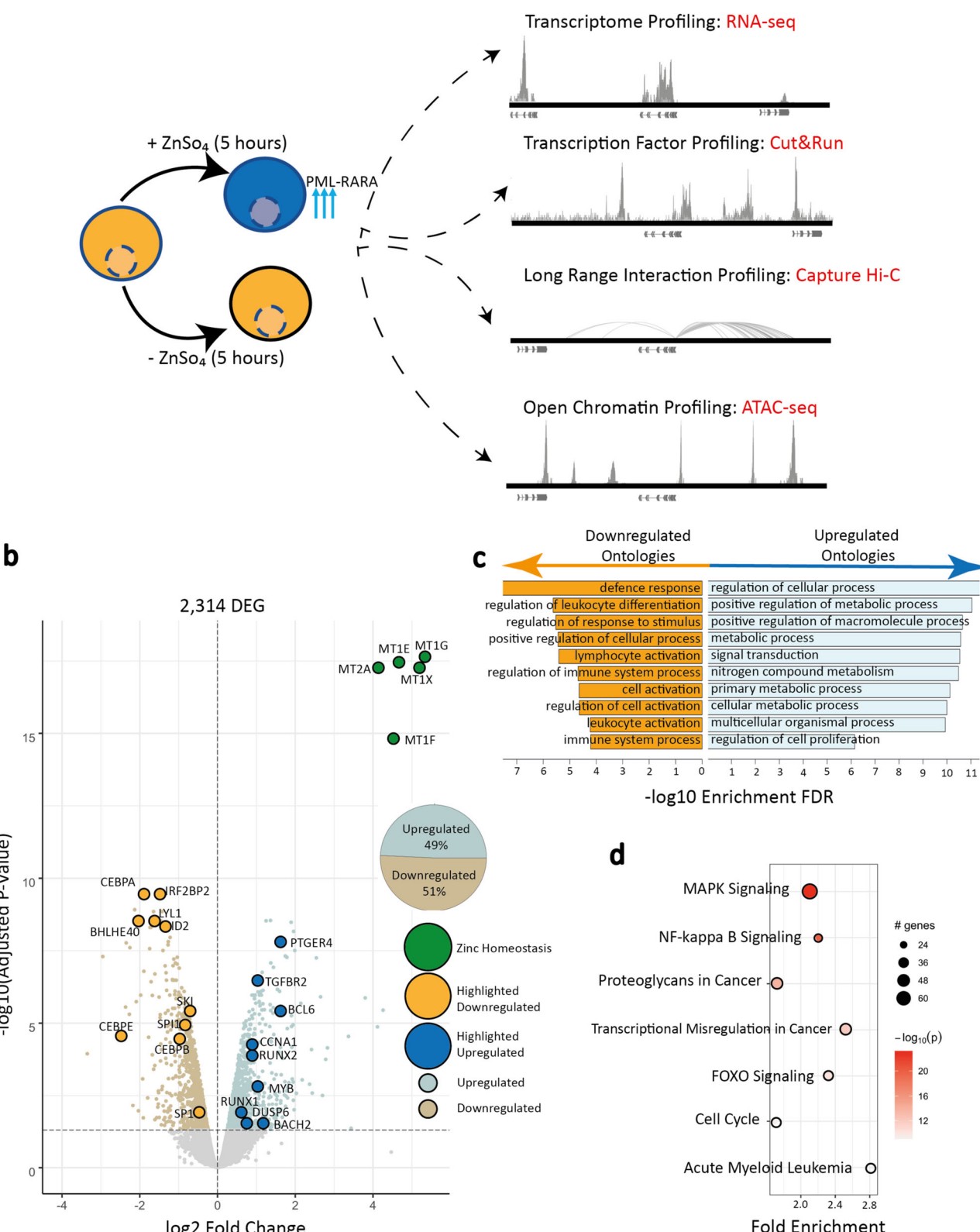

relaxation of the stringency only minimally increased the number of PML::RARA bound DEGs (Supplementary Fig. 2o). We contemplated that the 5-h zinc induction may not provide sufficient time for the steady-state mRNA levels of some genes to change. We drew comparisons of the NB4 PML::RARA binding profile with the U937-PR9 cells, as NB4 cells contain constitutively active PML::RARA and had

consistent binding(Supplementary Fig. 2l)[15]. Using publicly available NB4 RNA-seq[18], we compared expression levels between the two cell lines, focusing only on the genes that were firstly bound by PML::RARA in induced U937-PR9 cells yet did not significantly change expression upon induction, and secondly, are also bound by PML::RARA in the NB4 cell line (5311 gene promoters; Supplementary Fig. 2l). This

**Fig. 1 | PML::RARA induction instigates expression changes for thousands of genes. a** Schematic depicting the cell line model and datasets collected in this study. U937-PR9 cells (gold) have a zinc inducible promoter at a PML::RARA fusion gene, addition of ZnSO₄ to these cells for five hours leads to high expression of PML::RARA (light blue cell). For each induced and uninduced experiment, RNA-seq, Cut&Run, Capture Hi-C, and ATAC-seq libraries were made. **b** Volcano plot displaying the 2314 significantly DEGs, based on an FDR adjusted *p*-value cut off < = 0.05, showing downregulated (gold) and upregulated (blue) genes. Genes implicated in zinc homeostasis are shown (green). Numbers of upregulated and downregulated genes are shown in the inset pie chart. Significant DEGs were determined using Limma's linear modeling. **c** Bar plot showing the gene ontology enrichment of the up (blue) and down (gold) regulated genes. Higher −log10 enrichment scores indicate greater ontology enrichments. **d** Pathway enrichment of all DEGs. The size of the dot represents the number of genes within the given pathway, and the shade of red represents the significance of the DEG enrichment for the given pathway (−log10 enrichment *p* value determined by pathfindR internal algorithm and adjusted using Bonferroni correction). DEG = Differentially Expressed Gene. Source data are provided as a Source data file.

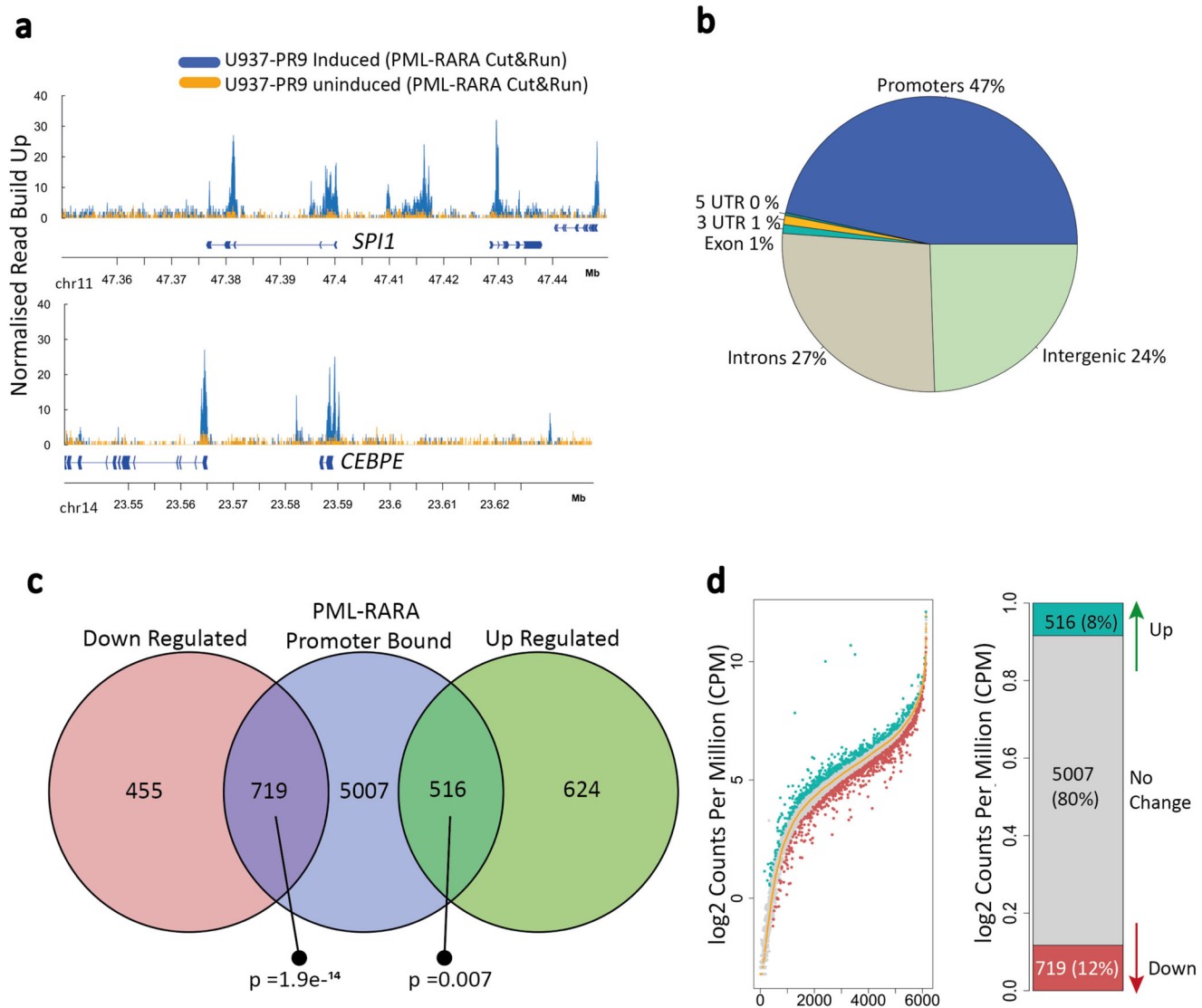

**Fig. 2 | Varying transcriptional responses of genes whose promoters bind PML::RARA. a** Genomic tracks of PML::RARA binding profiles at two key myeloid genes: *SPI1* (top) and *CEBPE* (bottom). Normalized PML::RARA read density in uninduced (orange) and induced (blue) U937-PR9 cells are shown. Blue indicates the replicate #1 PML::RARA induced profile and gold indicates the replicate #1 PML::RARA uninduced (background) profile. Y-axis represents the read count normalized by library size. The genomic coordinates are indicated along the x-axis. Gene bodies and exons are indicated by connecting blue blocks. **b** Pie chart showing the genomic distributions of all 15,412 PML::RARA peaks within promoters = blue, intergenic regions = light green, intronic regions = gray, exons = green and 3′ untranslated regions (3′ UTR) = orange. **c** Venn diagram overlapping the 6242 PML::RARA-promoter bound genes (blue) with upregulated (green) and downregulated (red) genes. The significance of each overlap is represented as the *p*-value using a χ² test. **d** Ranked plot showing the expression level distribution of all 6242 PML::RARA-bound genes in the PML::RARA-induced U937-PR9 cells, as compared to uninduced expression. Genes are ordered according to their expression level in uninduced cells. The expression of each gene in the uninduced cells is represented by the central orange line. The PML::RARA-induced expression of each gene is indicated by green, gray, and red dots indicating upregulation, no expressional change, and downregulation, respectively. Expressional change is based on genes having an adjusted *p*-value < = 0.05. The side bar plot shows the proportion of PML::RARA-bound genes that are upregulated, have no expressional change or downregulated. Source data are provided as a Source data file.

analysis showed no significant difference in expression levels for 85% of genes, suggesting that short exposure to the fusion protein in U937-PR9 cells is unlikely to account for a lack of response in the expression of these genes (Supplementary Fig. 2p, q). We also compared the level of gene expression in our RNA-seq datasets with those generated by others, who have recently carried out a time course of zinc induction in U937-PR9 cells[21]. Again, we noted that 59% of PML::RARA-bound genes that showed no change in our datasets were expressed at similar levels even after 24-h (Supplementary Fig. 2r). These results suggest that PML::RARA binding at promoters of many genes does not lead to a significant change in gene expression.

Secondly, we noted that 42% of the 1235 differentially expressed PML::RARA-bound genes were upregulated (Fig. 2c, d), consistent with recent findings[14]. Examination of promoter H3K9/K14 acetylation, a marker of gene activity, showed that levels are increased for upregulated genes, in contrast to the PML::RARA-bound genes whose expression decreased (Supplementary Fig. 2s). PML::RARA peaks were both stronger and wider at the promoters of up and downregulated genes, compared to those at stably expressed genes. (Supplementary Fig. 2t, u). We noted that downregulated genes had the strongest PML::RARA binding, however comparing the expressional change of down- and upregulated PML::RARA bound genes highlighted that the strength of binding did not consistently reflect a transcriptional change (Supplementary Fig. 2v). Together these data demonstrate an expanded and diverse functional consequence of PML::RARA binding at promoters.

### Long-range promoter interactions are gained and lost upon PML::RARA induction

Transcription is regulated through promoter interactions with distal elements such as enhancers and silencers and we contemplated that such engagements could modulate the role of PML::RARA recruitment at promoters. We applied promoter capture Hi-C to uninduced and zinc-induced U937-PR9 cells to map how genome-wide long-range regulatory interactions change with PML::RARA induction[22]. There were significant overlaps of interactions across all replicates, regardless of induction status, indicating high reproducibility of long-range interactions (Supplementary Fig. 3a, b). We pooled each replicate together and called significant interactions to create a consensus interaction set for each condition (Fig. 3a). We identified 185,494 and 211,582 statistically significant interactions in the uninduced and zinc-induced experiments, respectively, with on average 13.7 and 14.6 interactions detected per gene promoter (Fig. 3b). Generally, we noted a correlation between expression level and the number of interactions (Fig. 3c).

To focus on the changes to long-range contacts that occur with the induction of PML::RARA, we compared each uninduced and induced capture Hi-C library pairs to identify differential interactions. From these analyses, we identified 60,442 differential interactions that were present in at least two replicate experiments, which were split evenly between those that were gained in the induced state (30,039) and those that were lost (30,403) and were distributed across 10,910 genes (Supplementary Fig. 3c, d). Each promoter had a median of two differential interactions (Supplementary Fig. 3e). The multiple differential interactions of a gene were likely to be uniformly gained or lost; only 11% of genes contained both gained and lost interactions (Fig. 3d). This uniform pattern suggests that few genes appear to undergo 'rewiring' of contacts, replacing some contacts with others.

Integration of the differential interaction data with gene expression analysis revealed that 60% of differentially expressed genes engaged in differential interactions (Fig. 3e, f). By partitioning the data across the gained/lost and upregulated/downregulated axes, we observed that genes with gained interactions were twice more likely to be upregulated than downregulated (Fig. 3e, g). Similarly, loss of interactions was twice more likely to be associated with a loss of gene

expression (Fig. 3f, g). In contrast, genes with no change in expression did not show any bias in gained versus lost differential interactions (Supplementary Fig. 3f). Moreover, there was a greater enrichment for H3K9/K14ac, a marker of active enhancers[23], at the gained interacting partners of upregulated genes and lost interacting partners of downregulated genes, compared to elements involved in stable interactions (Fig. 3h). These results suggest that promoter engagement/disengagement with enhancers, rather than silencers, might be a stronger driver of expression patterns.

Next, we integrated the promoter interaction dataset with PML::RARA binding and noted that 74% of PML::RARA-occupied sites were positioned in fragments that engage in interactions in uninduced and/or induced cells (Supplementary Fig. 3g). Of these, 60% were differentially interacting, split evenly between interaction gains and losses (Supplementary Fig. 3h, i). It suggests that induction of PML::RARA leads to significant alterations in regulatory interactions with divergent consequences (Fig. 3i, j).

### Transcription factor motif analysis shows no distinctive enrichment pattern

Our results showed that the transcriptional consequence of PML::RARA binding at promoters was variable, yet this did not correlate with defined patterns of long-range interaction. We proposed that the local transcription factor binding site environment may be instrumental in directing the transcriptional outcome and aimed to examine the binding site environment at promoters of PML::RARA-bound genes and their long-range interacting partners. Firstly, to focus the analysis on sequences within fragments with regulatory potential, we generated ATAC-seq libraries to map regions accessible to transcription factors across the genome in uninduced and induced cells and identified 90,332 regions across our datasets (Supplementary Fig. 4a), distributed predominantly at promoters (18%), introns (35%) and intergenic regions (43%) (Supplementary Fig. 4b). We next carried out a differential chromatin accessibility analysis, examining the difference of read deposition between paired uninduced and zinc-induced samples at ATAC-seq peaks and identified 6376 significant differences (padj < = 0.1), with the vast majority (>99%) decreasing in chromatin accessibility upon zinc induction (Fig. 4a), although peaks were rarely completely extinguished (Fig. 4b). ATAC-seq peaks in promoter regions were particularly prone to change upon induction (Supplementary Fig. 4b, c). In all, 3596 gene promoters were associated with one or more closing ATAC-seq peaks. Over 97% of PML::RARA peaks were localized centrally at pre-existing ATAC-seq peaks, suggesting that the fusion protein is recruited to sites where other factors are already bound (Fig. 4c and Supplementary Fig. 4d). ATAC-seq peaks that underwent a peak size reduction upon PML::RARA induction were more likely to be associated with stronger PML::RARA binding than stable ATAC-seq peaks (Supplementary Fig. 4e). We considered that differences in ATAC-seq peak stability at PML::RARA binding promoters may be reflected in different gene expression responses. While not entirely correlated, genes with PML::RARA bound promoters and differential ATAC-seq peaks were more likely to be downregulated than upregulated (Supplementary Fig. 4f). Differentially expressed genes with PML::RARA-bound promoters and stable ATAC-seq peaks did not display this skew (Supplementary Fig. 4g). Examination of the H3K9/K14ac signal at these sites revealed that genes with increased expression also gain promoter acetylation, in contrast to genes with decreased expression, which typically undergo no such change (Supplementary Fig. 4h). These results corroborate that promoter recruitment of PML::RARA does not uniformly lead to a loss of acetylation and expression, as is widely reported. It implies that other, contextual factors have an impact on the transcription output.

Next, we segregated the promoters and interacting partner elements associated with PML::RARA binding into six categories, based

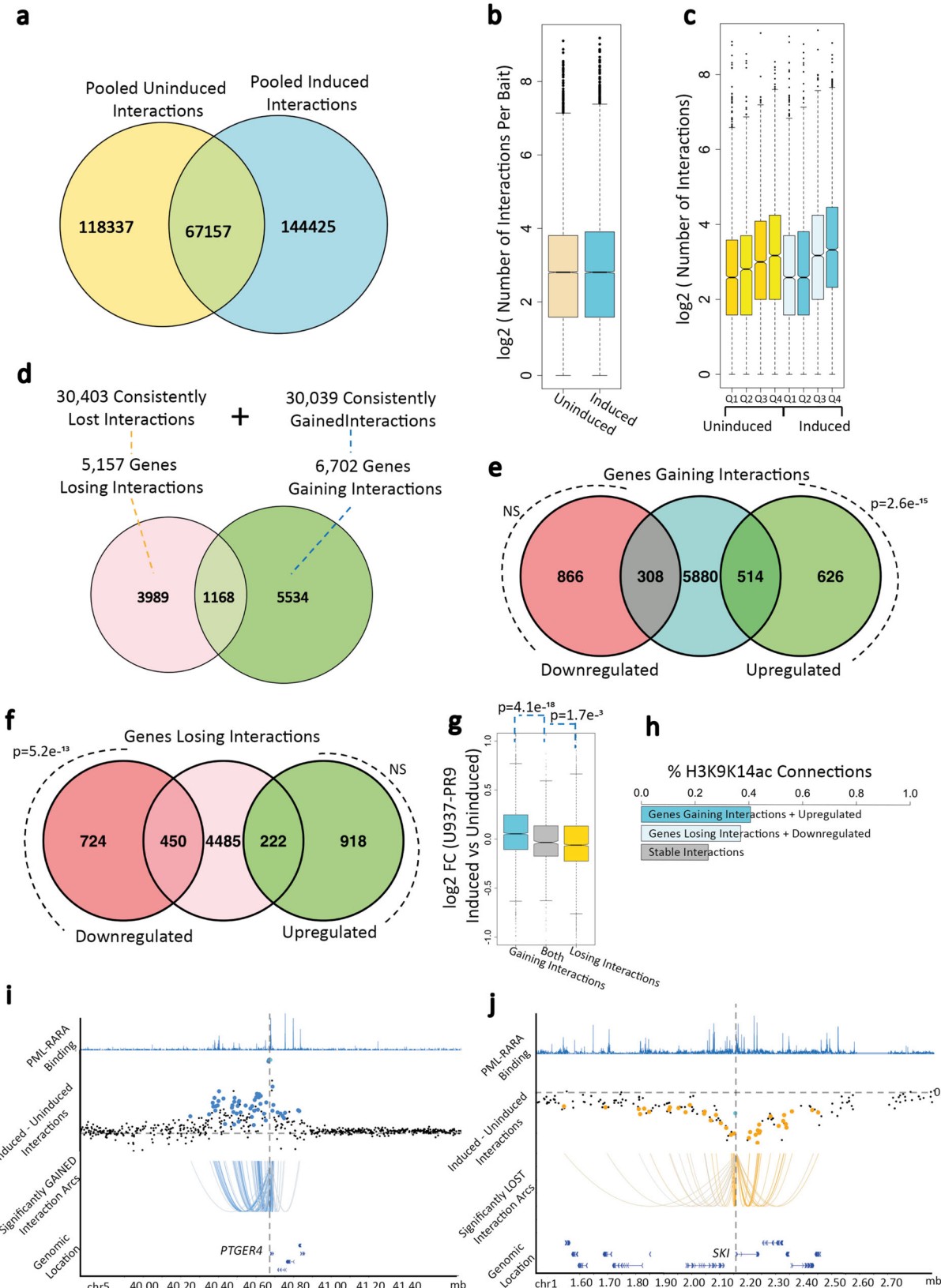

upon their transcriptional response (up- and downregulated and no change) and long-range interaction changes (gained and lost), and carried out differential motif searches, focusing on 400 bp surrounding the ATAC-seq peak apex (Fig. 5a). In each category, we found a subtle enrichment for certain transcription factors compared to elements within other groups, however, it accounted for only a fraction of

the elements (Supplementary Fig. 5a). We did not identify any single TF that was consistently associated exclusively with a single transcription/ long-range interaction category.

We also compared the collective transcription factor binding site (TFBS) composition of each element across all categories to determine whether there are consistent similarities that segregate the different

**Fig. 3 | Gains and losses of long-range interactions upon PML::RARA induction. a** Overlap of significant (CHiCAGO score>5) interactions identified in PML::RARA-induced (cyan) and uninduced (orange) U937-PR9 cells. **b** The number of significant interactions per baited promoter across uninduced (orange) and PML::RARA induced (cyan) interactions (from three biological replicates across 19,023 genes). Y-axis is log2 scale. **c** The number of significant interactions per gene (from three biological replicates across 19,023 genes), separated into expression quartiles. Q1 = genes with log2CPM < −5.12, Q2 = genes with log2CPM expression ranging from −5.12 to 1.7, Q3 = genes with log2CPM expression ranging from 1.7 to 5.3, Q4 = genes with log2CPM expression >5.3. The first four boxes refer to interactions and expression levels in the uninduced cells (orange) and the second four boxes refer to those in PML::RARA-induced cells (blue). **d** Overlap of genes that gain interactions (green) and genes that lose interactions (pink). **e** Overlap of genes that gain interactions (teal) and genes whose expression is significantly increased (green) or decreased (red). P refers to the significance of the overlap as determined by $\chi^2$ test. NS = not significant. **f** Overlap of genes that lose interactions (pink) and genes whose expression is significantly increased (green) or decreased (red). P refers to the significance of the overlap as determined by $\chi^2$ test. NS = not significant. **g** Log2 fold expression change of induced vs uninduced cells for genes that exclusively gain interactions (cyan), both gain and lose interactions (gray), and

exclusively lose interactions (orange). p represents the two-sided t-test p-value comparing the mean expression levels of the different categories. N = two RNA-seq biological replicates. **h** Percentage of interacting non-promoter ends that overlap with H3K9/K14ac ChIP-seq peaks. **i** PML::RARA binding sites and gained interaction profiles at the *PTGER4* gene, which is upregulated upon PML::RARA induction. The top row shows the genes and their genomic coordinates. Second panel shows the PML::RARA binding profile. Third panel shows capture Hi-C interaction read count difference of induced vs uninduced cells. Each dot represents a HindIII fragment. Enlarged dots indicate that the HindIII fragment has a significantly greater number of reads in the PML::RARA induced sample and is considered a gained interaction. The fourth panel shows an arc plot, connecting the *PTGER4* gene with all HindIII fragments with significantly differential interactions. **j** Similar landscape plot as (**i**) but representing the *SKI* gene locus which loses interactions and is down-regulated, indicated by orange. All boxplots were plotted with identical parameters: minima and maxima are indicated either as the lowest and highest outliers or as the lower and upper whiskers if the values are within −1.5 and 1.5x the interquartile range from the lower and upper quartiles, respectively. The box bounds correspond to the first and third quartiles. The center indicates the median−all outliers are plotted. Source data are provided as a Source data file.

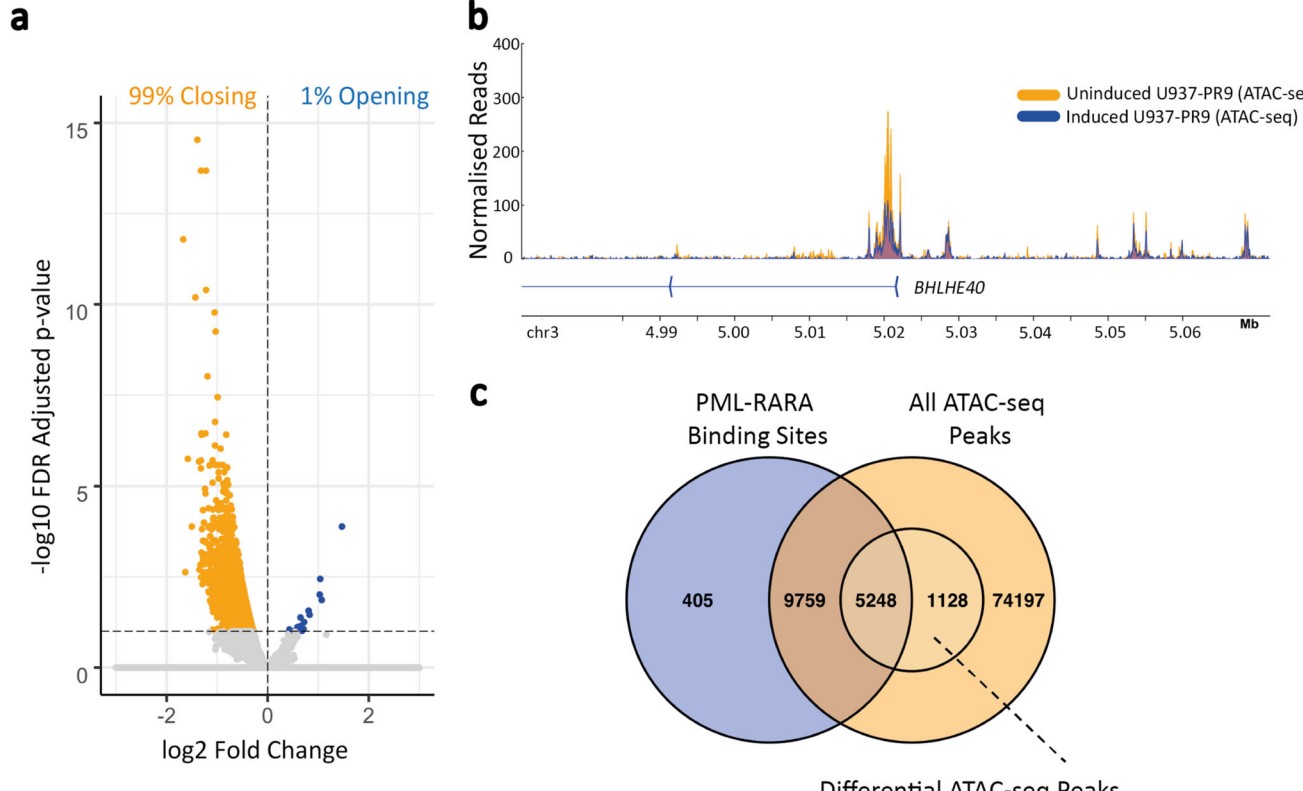

**Fig. 4 | PML::RARA expression induces widespread reduction of open chromatin regions. a** Volcano plot showing the differential ATAC-seq peaks comparing PML::RARA-induced cells to uninduced cells. Each dot represents an ATAC-seq peak. Peaks significantly reduced in size are colored orange and significantly increasing in size are blue. Significance threshold is adjusted p-value < = 0.1. **b** Landscape plot showing an example of a differential 'closing' ATAC-seq peak at

the *BHLHE40* promoter. ATAC read coverage profile in uninduced (orange) and induced (blue) replicate #1 samples are shown. Reads are normalized by library size. **c** Venn diagram showing overlap of the 15,412 PML::RARA binding sites with the 90,322 ATAC-seq peaks and the subset of 6376 closing ATAC-seq peaks. Source data are provided as a Source data file.

classes of elements. Visualization by tSNE showed there was effectively no distinction between the composition of motifs across the different categories (Fig. 5b). This suggested that either the local motif environment at PML::RARA-associated elements has little contribution in directing its role, or that the patterns are overly complex to observe through a general motif comparison.

## Machine learning algorithms identify distinct TF motif compositions that define responses to PML::RARA binding

Machine learning algorithms are well suited for the integration of large and complex NGS datasets[24]. In particular, the eXtreme Gradient Boosting (XGBoost) model has shown promise at deconvoluting complex patterns with high interpretability[25]. We aimed to train

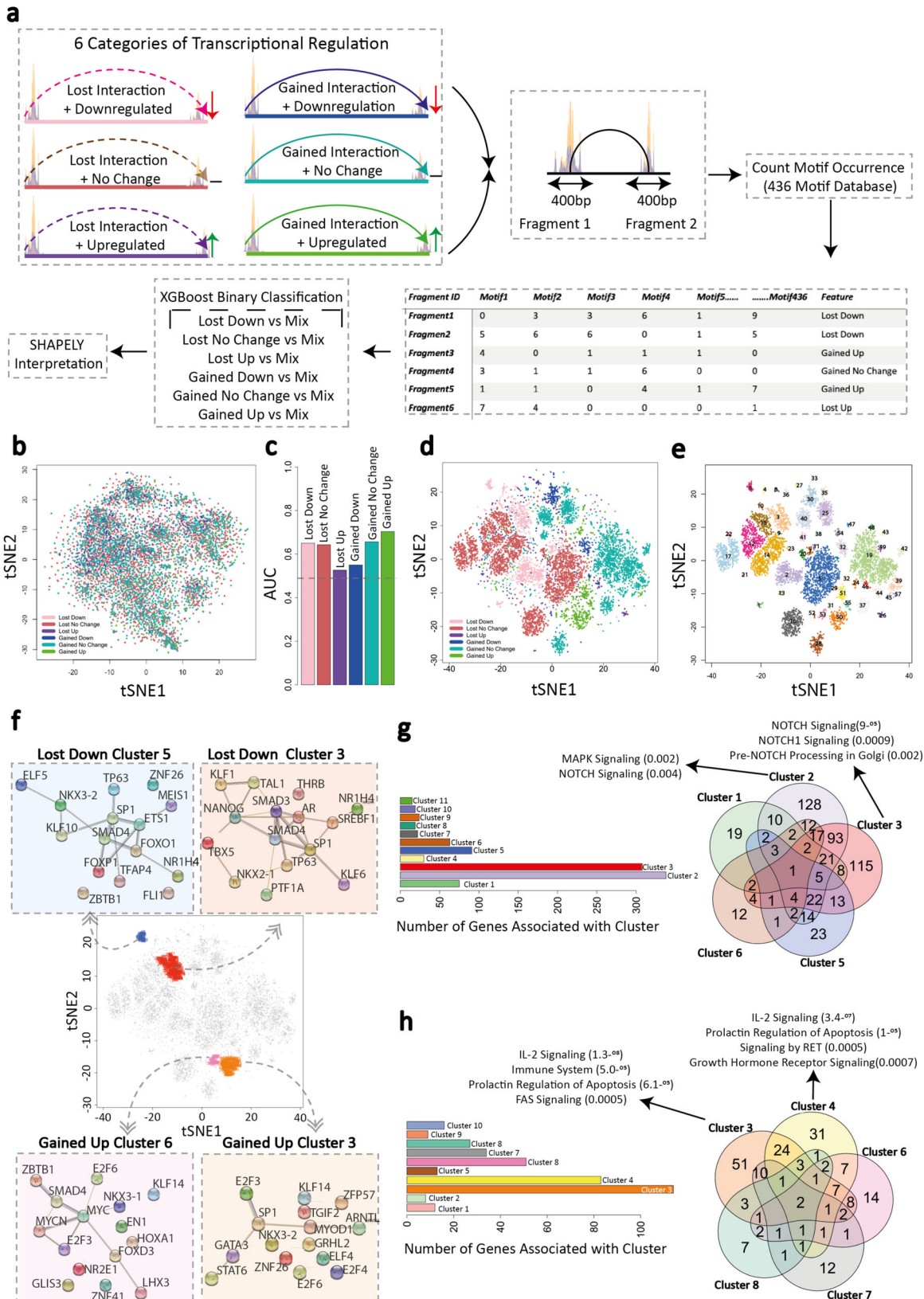

XGBoost models to predict the regulatory category a genomic element is involved in based on its TFBS composition (Fig. 5a). We trained each model on 80% of the dataset, then set it to predict the remaining 20% of data. Initially, models were trained using pairwise comparisons across all categories (one vs one) and attained predictive scores reaching an Area Under the Curve (AUC) score of 0.79, indicating that

the elemental motif landscape can be highly predictive (Supplementary Fig. 5b). We subsequently generated models that compared each category to an equal-sized mixture of all other categories (one vs all), to determine if a general distinctive pattern could be learnt. Predictive scores were lower than for the pairwise comparisons, ranging from AUC scores of 0.55 to 0.70. Our results suggest that patterns can be

**Fig. 5 | Machine learning algorithms identify distinct TF motif compositions that define responses to PML::RARA binding. a** Classification of PML::RARA-associated fragments into six gene interaction/expression categories. Transcription factor motif analysis for both interacting fragments is carried out across 400 bp in the center of the ATAC-seq peaks. The motif composition for each fragment is used to develop machine learning models. **b** tSNE plot visualizing the clustering of ATAC-seq peaks based on motif composition similarities without machine learning interpretation. Each dot represents an ATAC-seq peak and is colored according to the interaction/expression category to which the peak-containing fragment belongs (pink, lost/downregulated; brown, lost/no change; purple, lost/upregulated; blue, gained/downregulated; teal, gained/no change; green, gained/upregulated). **c** Bar plot showing the AUC scores for each one-vs-all machine learning model. Dashed line (0.5) represents random predictions, and scores above that indicate predictive power. AUC = Area Under the Curve. **d** tSNE plot visualizing clustering of ATAC-seq peaks based on the SHAPELY weighting derived from machine learning. Each dot is colored according to the interaction/ expression category to which the fragment containing the ATAC-seq peak belongs. **e** Identification of ATAC-seq peak clusters based on the SHAPELY weighting scores

derived from machine learning. Each cluster is assigned a color using DBSCAN identified clusters; those points not assigned to a cluster are removed from the plot. **f** Protein–protein Interaction network of TFs, which bind the top 15 predictive motifs in a downregulated/lost interaction cluster (#5 and #3, blue and red, respectively) and in an upregulated/gained interaction cluster (#6 and #3, pink and orange, respectively). Circles are TFs and lines show known and predicted physical interactions between them as identified by the STRING database. PML and RARA are denoted as stars. Each plot centers around the tSNE plot highlighting clusters to which each network refers. **g** Bar plot (left) indicating the numbers of genes associated with each of the 11 downregulated/lost interaction clusters identified from (**e**). Venn diagram (right) showing the overlap of genes within the five largest clusters. For two clusters (#3 and #4), the top three enriched pathways are shown. **h** Bar plot (left) indicating the numbers of genes associated with each of the ten upregulated/gained interaction clusters identified from (**e**). Venn diagram (right) showing the overlap of genes within the five largest clusters. For two clusters (#3 and #4), the top three enriched pathways are shown. Source data are provided as a Source data file.

learned and highlight that TFBS signatures can uniquely define a category (Fig. 5c).

To identify the TFBSs that facilitated the model to correctly predict an outcome, we applied the SHapely Additive Explanations (SHAPELY) algorithm to each one-vs-all model[26]. SHAPELY calculates the weighted contributions of predictive features for each element, giving a score for each TFBS. We re-examined the t-SNE representation in Fig. 5b, this time supplying the SHAPELY weighting scores, rather than the absolute number of each TFBS. In contrast to the mixed pattern that was observed without machine learning, we found a strong separation of each transcription/interaction category, with little overlap (Fig. 5d).

We next classified 57 spatially distinct clusters, ranging from three to seventeen clusters per transcription/interaction category (Fig. 5e). Interrogation of the top defining drivers in each cluster revealed that either the presence or absence of a TFBS could be predictive (Supplementary Fig. 5c). On average, the highest contributing TFBS accounted for only 10% of the prediction, suggesting that the combination of features required are highly complex (Supplementary Fig. 5d). For example, the top five features driving the lost interaction/ downregulated cluster #3 predictions were motifs ZFX, ZAC1, ZNF71, ZNF41, and SREBP1 (Supplementary Fig. 5e). The presence of a ZFX binding site was the most contributing to the prediction, yet still only accounted for 6% of the prediction. Mapping of ZFX binding sites demonstrated a widely distributed pattern across all clusters (Supplementary Fig. 5f). The SHAPELY interpreted distribution of ZFX creates a focus on cluster #3, although still excludes 70% of the elements within the cluster as having a ZFX binding site (Supplementary Fig. 5g). It demonstrates that the simple presence or absence of a motif is insufficient to distinguish a cluster from others; the machine learnt pattern likely applies additional weight that considers factors such as combinatorial TFBS and numbers of motifs.

We next considered the relationships between the transcription factors that potentially bind the collection of motifs within a cluster. Focusing on the TFs that bind the top-15 present, SHAPELY weighted motifs within each cluster, we saw that each cluster contained binding sites for TFs that are known to interact, in some instances forming interconnected protein interaction networks (Fig. 5f). Such transcription factors within these elements may function cooperatively to exert transcriptional control.

Lastly, we examined commonalities that connect genes within a single cluster. Analyses revealed individual clusters displayed enrichments for specific signaling pathways. For instance, four clusters within the upregulated/ gained interaction category were enriched for genes that are the targets for IL-2 signaling (Fig. 5g). Within the downregulated/lost interaction category, separate clusters were

enriched for genes involved in NOTCH, MAPK, and MYC signaling pathways (Fig. 5h). These observations are indicative of shared regulatory programmes of genes that may require coordinated expression.

## Characterization of expression and long-range interaction patterns in APL patients

Finally, we considered whether the inducible U937-PR9 cell line system can be a suitable model for primary APL by comparing its patterns of long-range interaction and gene expression to data collected from two patients harboring PML::RARA rearrangements. The global gene expression profiles between the two patients exhibited a high degree of correlation ($r = 0.91$), and correlated highly with the PML::RARA induced U937-PR9 cells ($r = 0.85/0.82$, Patient #1, Patient #2) (Fig. 6a and Supplementary Fig. 6a). Restricted to genes whose promoters bound PML::RARA, there was a drop in correlation between patients ($r = 0.80$) and between the patients and the cell line ($r = 0.74,0.67$, Patient #1, Patient #2), indicating that expression patterns of PML::RARA bound genes are more prone to divergence (Supplementary Fig. 6a, b).

We next compared the patterns of long-range interactions in the patients and induced cell line. There was a significant overlap across all interaction profiles, with a 61 to 75% overlap between the two patients and a 52 to 49% overlap for patient #1 and patient #2 compared against the cell line, respectively (Supplementary Fig. 6c–e). We observed that 2956 of the gained interactions identified in the U937-PR9 cell line model were established interactions in both patients, which was significantly greater than expected by chance (Fig. 6b). The overlapping gained interactions involved 2421 genes, most of which showed a very strong expression correlation between the patients and cell lines ($r = 0.87$) (Fig. 6c, d). Highly correlative genes included many that have roles in cell proliferation, including *PTGER4, VIM, CCNA1, AHR, TGFBR2, MYB, MYBL1, DUSP6, CHD2, PAG1, FYN,* and *NFAT5* (Fig. 6e–j and Supplementary Fig. 6f), all of which were upregulated after PML::RARA binding. These comparisons highlight that genes upregulated by PML::RARA-dependent chromatin remodeling remain engaged and highly expressed in patients.

Generally, we observed that genes with the highest degree of expression similarity between the induced cell line and the patient samples were significantly more likely to exhibit similar long-range interactions than those with the lowest concordance (Supplementary Fig. 6g–l). There were however examples where the expression levels and long-range interaction profile similarities were recapitulated in only one of the two patient samples (Supplementary Fig. 6k). We also identified genes that had discordant gene expression levels across patients and cell line yet exhibited highly similar

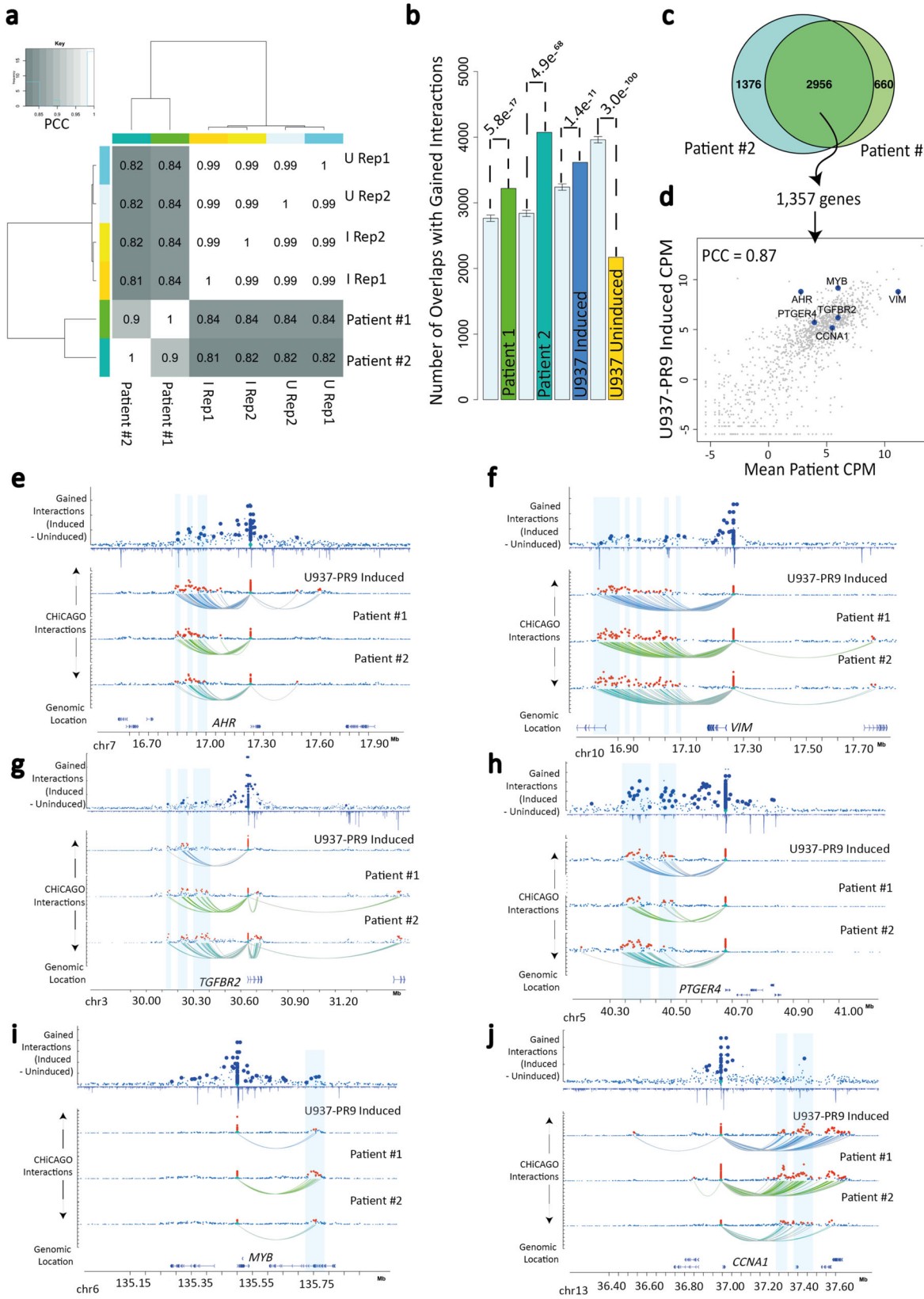

interaction profiles (Supplementary Fig. 6l). Taken together, our results indicate that many but not all expression and long-range interaction profiles are matched between the cell line model and primary patient samples. Similar expression outputs are frequently a predictor of shared regulatory inputs, although this relationship is not universal.

Lastly, we used the long-range interactions and transcription datasets collected from the two APL patients, supplemented with ATAC-seq datasets derived from additional patients, to generate machine-learnt models in comparison with publicly available datasets produced from CD34 + haematopoietic progenitor cells derived from healthy individuals. Following training executed under the same

**Fig. 6 | Patterns of expression and long-range interactions in APL patients.**
**a** Heatmap showing the Pearson correlation coefficients of global gene expression across the two APL patients and the induced/uninduced U937-PR9 cell line replicates. **b** Bar plot showing the number of regions overlapping with the 30,039 gained interactions identified in the U937-PR9 cell line model. The light blue bar illustrates the expected overlap of the gained regions as expected by chance, error bars indicate the range of 20,000 Monte Carlo Simulations. P indicates a significantly greater number of overlaps by chance, as determined by $\chi^2$ test. The data used to subsample consisted of five independent interaction datasets–for each simulation a random set of interactions of the same size as the comparison set was taken. Mean error bars for each set of simulations are 2750, 2810, 3246, and 3980 for Patient #1, Patient #2, induced U937, and uninduced U937 datasets, respectively. **c** Venn diagram showing the overlap of interactions in the patient samples that correspond to gained interactions in the cell line model. The intersection

contains 2956 fragments that interact with 1357 genes. **d** Scatter plot showing the mean expression correlation of the 1357 genes, comparing patients and induced U937-PR9 cells. Pearson correlation coefficient = 0.87. Positions of six genes relating to (**e**) are shown (blue dots). **e–j** Interaction landscape plots of six genes: *AHR*, *VIM*, *TGFBR2*, *PTGER4*, *MYB*, and *CCNA1*. The top panel in each plot shows the gained interactions after PML::RARA induction in the U937-PR9 cells, large blue dots indicate regions with a significant differential (gained) interaction, as determined by GOTHiC (ihw < = 0.01). The blue inverted peak tracks show the binding profiles of PML::RARA in U937-PR9 cells. Arc plots show the interactions for the induced U937-PR9 cells (blue), Patient #1 (light green), and Patient #2 (dark green). The colored arcs connect significant interactions to the gene promoter. Significant interactions are displayed as a red dot, and non-significant interactions as a blue dot. Each dot represents a unique HindIII fragment. Source data are provided as a Source data file.

conditions as modeling in the U937-PR9 cells, we generated models that could correctly predict the activities and behaviors of interacting fragments within different functional outcome classes, with AUC ranging between 0.51 and 0.64 (Fig. 7a, b). Clustering based upon the associated SHAPELY weighting demonstrated an effective segregation between clusters of functional categories of elements (Fig. 7c, d). Consistent with the U937-PR9 modeling, the most influential TFBS based upon the SHAPELY weighting showed evidence of physical interactions, and the associated genes within the clusters demonstrated functional enrichments that are relevant to both CD34 + progenitor (e.g., stem cell maintenance) and APL biology (e.g., neutrophils and antigen processing/presentation) (Fig. 7e–h). These analyses demonstrate a wider applicability of machine-led model generation, beyond the cell line system, to infer functionality of regulatory elements across other experimental systems.

## Discussion

In this study, we have applied a multi-omics strategy to characterize the gene expression and regulation consequences of fusion protein formation in APL, using an inducible cell line model. While other groups previously have used overlapping PML and RARA binding profiles from ChIP-seq and ChIP-chip experiments to characterize PML::RARA fusion protein recruitment, our approach employs a PML::RARA fusion-specific antibody using Cut&Run. It delivers a considerably more comprehensive dataset, which is also paired with gene expression and long-range interaction analysis. A recent study also examined the impact of PML::RARA binding on long-range regulatory interactions, although it lacked experimental replicates, thereby confounding interpretability[21]. Our study design is supported by interrelated, replicated datasets that support statistically validated analysis.

The most focus of PML::RARA action in APL centers on its contribution to a myeloid differentiation block, however, the aggressiveness of the disease stems from heightened proliferation. Our findings suggest that PML::RARA may play a direct role in driving these events. Strikingly, over 40% of differentially expressed, PML::RARA-bound genes were upregulated. Promoter recruitment of PML::RARA has previously been associated with increased transcription of the *MYB* gene[13]. During the preparation of this manuscript, a report determined that PML::RARA association is required for high expression levels across hundreds of genes in NB4 cells[14]. Our findings support this observation and demonstrate that this phenomenon is widespread and include many genes with cell proliferation functions. Equally notable was that 80% of PML::RARA-recruiting genes do not significantly alter their expression output, which implies that PML::RARA requires a conducive binding environment to exert any transcriptional effect, or potentially additional signaling inputs that are not provided in our experimental system.

Integration of the multi-omics datasets reveals gene targets of PML::RARA that are likely to have clinical implications for APL. For instance, in the U937-PR9 cell line we identified several coagulation-

associated genes (*ANXA2*, *ANXA7*, *VIM*, *F3*, *IL1B*, and *IL1RAP*) as upregulated binding targets of PML::RARA, which was corroborated by patient sample RNA-seq and preliminary PML::RARA binding profiling by CUT&Tag. The high expression of these genes is directly implicated in the disruption of coagulation cascades in disease[27]. Coagulopathy and hemorrhagic events still pose major challenges for APL in the clinic[28]. Our data support a hypothesis that such risks could arise through intrinsic properties within blast cells, through PML::RARA dependant mechanisms[29].

Divergent consequences of PML::RARA binding imply that different mechanisms are invoked. We noted that there was a general bias toward upregulated genes gaining long-range interactions, and downregulated genes losing interactions, which implies that enhancer engagement and disengagement plays a role in modulating PML::RARA induced expression changes. Yet a significant minority of genes lose interactions with distal elements with increased expression or gain interactions with decreased expression, which may reflect the participation of repressing elements such as silencers in some cases.

PML::RARA is recruited to pre-existing ATAC-seq peaks, which suggests that it moves to sites that are already occupied by other transcription factors. It is not clear whether it binds in addition to other factors or displaces them. A third of PML::RARA binding sites occur at ATAC-seq peaks that undergo a size reduction but do not extinguish completely. Peak reconfiguration does not map on to a specific transcriptional outcome, nor is it associated with a particular transcription factor binding site signature. While ATAC-seq peaks are typically considered a marker of gene activity, an alteration to peak intensity may sometimes reflect the changing composition of the resident factors without an obligatory change in expression.

Since the divergent transcriptional outcomes of PML::RARA binding could not be distinguished based on a specified pattern of long-range interaction or chromatin occupancy changes, nor singly enriched transcription factor binding sites, we applied the machine learning strategy to uncover convoluted patterns of defining transcription factor binding sites. REBEL uncovered clear discrimination of TFBS patterns within promoters and distal interacting regions into multiple sub-clusters for each functional outcome. Many of the most highly predictive TFBS such as SP1, MYB, and SMAD4 have been implicated in APL and appear to influence PML::RARA behavior through intricate combinations[30–32]. Both promoters and distal interacting elements populate and are only marginally segregated within the sub-clusters, implying that the machine is unable to fully discern these types of elements by TFBS composition. It is remarkable how complex is the pattern needed to cluster elements. Permutations are difficult to detect by eye but clearly can highlight some defined TF interaction networks and signaling pathways. Particularly for the modeling based upon CD34 + cell and APL patient blast cell comparisons, it emphasized both transcription factors and gene ontology functional enrichments that are meaningful for those cell populations and can contribute to further biological understandings. We did not

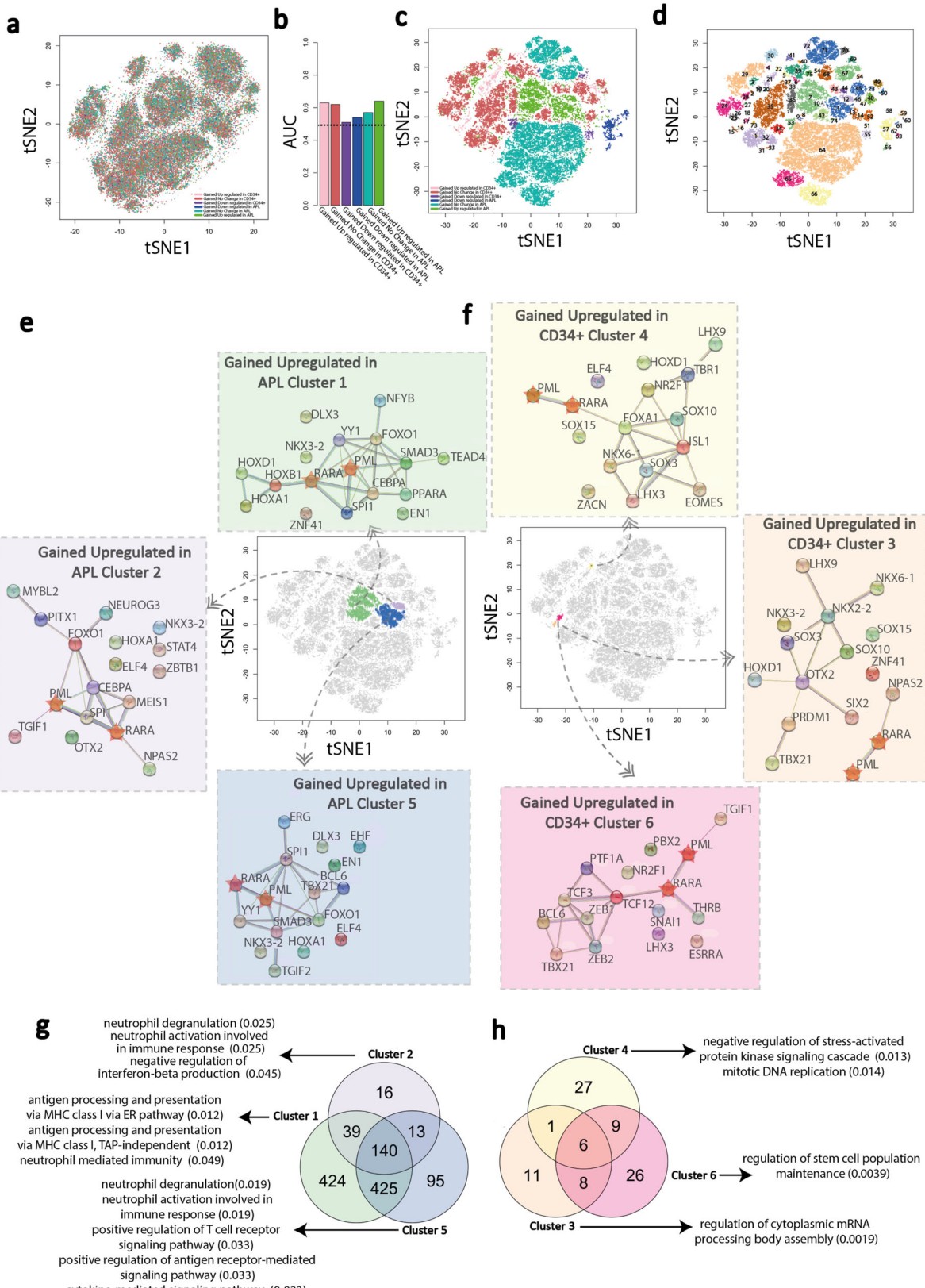

include relative TFBS spatial information into our model, due to the limited dataset size on which to train such a complex model. Its inclusion in an expanded study may further refine patterns. The richness of NGS datasets has to date been underexploited. With the wealth of publicly available datasets, these methods can be widely applied across experimental systems and cell types to comprehend complex transcriptional responses.

Cell lines are used widely to study disease, but it is often undetermined whether they faithfully recapitulate those that they are meant to model. Our study indicates that the gene regulation

**Fig. 7 | Interrogation of the machine learning architecture applied to the APL patient vs CD34 + model. a** tSNE plot visualizing the clustering of ATAC-seq peaks based on motif composition similarities without machine learning interpretation. Each dot represents an ATAC-seq peak derived from CD34 + and APL patient samples and are colored according to the differential interaction/expression category to which the peak-containing fragment belongs: (pink, gained/upregulated in CD34 + cells; brown, gained/no change in CD34 + cells; purple, gained/downregulated in CD34 + cells; blue, gained/downregulated in APL patients; teal, gained/no change in APL patients; green, gained/upregulated in APL patients. **b** Bar plot showing the AUC scores for each APL patient vs CD34 + machine learning model. Dashed line (0.5) represents random predictions, and scores above that indicate predictive power. AUC = area under the curve. **c** tSNE plot visualizing clustering of ATAC-seq peaks based on the SHAPELY weighting derived from the APL patient vs CD34 + machine learning models. Each dot is colored according to the differential interaction/expression category to which the fragment containing the ATAC-seq peak belongs. **d** Identification of ATAC-seq peak clusters based on the SHAPELY weighting derived from machine learning. Each cluster is assigned a

unique color using DBSCAN identified clusters; those points not assigned to a cluster are removed from the plot. **e, f** Protein–protein interaction network of TFs, which bind the top-15 predictive motifs in the gained/upregulated in APL patients cluster (#1, #2, and #5, green, purple, and blue, respectively) and in gained/upregulated in CD34 + cells cluster (#3, #4, and #6, orange, yellow, and pink, respectively). Circles are TFs and lines show known and predicted physical interactions between them, as identified by the STRING database. PML and RARA are denoted as stars. Each plot centers around the tSNE plot highlighting clusters to which each network refers. **g, h** Three-way Venn diagrams showing the overlap of genes associated with the fragments within the three clusters depicted in (**e, f**) for the gained/upregulated in APL patients cluster (#1, #2, and #5, green, purple, and blue, respectively) and in gained/upregulated in CD34 + cells cluster (#3, #4, and #6, orange, yellow, and pink, respectively). The top enriched gene ontologies—determined using the EnrichR online tools reported enrichments—associated with each gene set is highlighted with an arrow—only those with an adjusted $p$ value <= 0.05 are shown. Source data are provided as a Source data file.

and expression landscape for many genes in U937-PR9 cells are largely preserved in patients. Yet discrepancies do exist, so a degree of caution is advisable. The complicated karyotype and mutational burden of U937-PR9 cells is likely to underlie these differences. Largely, expression output equivalence is a strong predictor for corresponding long-range contacts, but this is not universal. Therefore, individual assessment of expression and interaction is needed on a gene-by-gene basis to gauge the suitability of the cell line as a model. It's notable that expression and long-range interaction divergences exist between the patients for a small number of genes. Such discrepancies between individuals may contribute to different responses to treatments and may ultimately assist the stratification of disease.

# Methods

## Patient samples
Patient samples were collected from the Haematology Unit, Guy's Hospital, London, UK. Each sample contained >90% blast cells and were used for RNA extraction and capture Hi-C library preparation. Samples of fresh peripheral blood were donated at diagnosis with written informed consent in accordance with the Declaration of Helsinki and with approval from the London–Westminster Research Ethics Committee. (IRAS project ID: 220344, REX reference 06/Q0702/140).

## Cell culture
U937-PR9 and NB4 cells were cultured in RPMI 1640 (Thermo Fisher, A10491), supplemented with 10% fetal bovine serum (Thermo Fisher, 10082147), and incubated at 37 °C, 5% $CO_2$. For the zinc induction of U937-PR9 cells, $ZnSO_4$ (Sigma, Z0251) was dissolved in sterile water as a stock solution at 100 mM. $ZnSO_4$ was added directly to the culture medium at a final concentration of 100 μM and incubated at 37 °C for 5 h, as described previously[15]. For the non-induced U937-PR9 cells, sterile water was added instead of $ZnSO_4$ and incubated under the same conditions.

## RNA extraction and RNA-seq library preparation
RNA was extracted from U937-PR9 cells using Trizol reagent (Life Technologies). Libraries were prepared using the Illumina SureSelect Strand-Specific RNA Library Prep kit following the manufacturer's instructions and sequenced using an Illumina HiSeq 2500 instrument for paired-end sequencing.

## Software parameters
For most data processing default parameters were used, unless stated otherwise.

## RNA-seq analysis
Reads were trimmed of adapters and low-quality reads using Trimmomatic[33] and aligned to the hg19 genome using STAR 2-pass aligner as outlined previously[34]. Gene counts were generated using Feature Counts from the Rsubread R package[35] and normalized using the voom normalization approach for differential analysis[36]. Differential analysis comparing uninduced and induced U937-PR9 cells was performed using limma[36]. Differentially expressed genes were considered if they had an adjusted (Benjamin Hochberg) $p$-value <= 0.05. For correlative analysis, log2 counts per million reads were used.

## PML::RARA fusion transcript quantitation
PML::RARA transcript levels within U937-PR9, APL patient, and publicly available NB4 RNA-seq datasets were quantified using Arriba[37]. Briefly, raw fastq files were aligned to the hg19 genome using the STAR aligner with custom parameters specific to retain chimeric reads. The chimeric BAM files were then used as input to Arriba using default parameters. The number of PML::RARA spanning reads for each sample were extracted and normalized by the number of total reads for that sample (chimeric transcripts/total reads × $1e^6$).

## Western blotting
Whole cell extracts were prepared with M-PER Mammalian Protein Extraction Reagent (Thermo Scientific) supplemented with protease inhibitors (Roche). Protein concentrations were determined using the Bradford assay (Bio-Rad Protein Assay) calibrated with a BSA standard curve. NuPAGE 4–12% Bis-Tris gradient polyacrylamide gels (Thermo Scientific) were used for SDS-PAGE. Blotting was performed using 0.2 μm nitrocellulose membrane (GE Healthcare). The primary antibodies used were: anti-RARA (Diagenode C15310155–same as for Cut&Run) 1:5000 dilution in 5% BSA, and anti-GAPDH (Merck Millipore CB1001) 1:10,000 dilution in 5% BSA. Either goat anti-rabbit IgG-HRP (Thermo Scientific) or goat anti-mouse IgG-DyLight 680 (Thermo Scientific) was used as the secondary antibody (1:5000 dilution). Chemiluminescent and fluorescent signal detection were performed using the iBright Imaging system (Thermo Scientific). Semi-quantitative analysis of blotting signal was performed using Image Lab (BioRad) software with PML::RARA protein expression normalized to GAPDH. Full blots are provided in the Supplementary Data.

## Hi-C library generation
Hi-C library generation was carried out as described previously[22]. For each condition and each replicate, 20 million cells were fixed in 2% formaldehyde for 5 min, cells were incubated on ice for 30 min in 25 mL of ice-cold lysis buffer. After overnight digestion with HindIII at 37 °C, DNA ends were labeled with biotin-14–dATP (Life Technologies)

using a Klenow end-filling reaction. Biotinylated DNA ends were then ligated together in an overnight ligation step using T4 DNA ligase (Invitrogen). After phenol: chloroform/ethanol purification DNA was quantified using Qubit, with a maximum of 40 µg taken forward. DNA was sheared to a peak concentration of around 400 bp, using the manufacturer's instructions (Covaris). Sheared DNA was then end-repaired, polyadenine tailed, and double size selected using AMPure XP beads to isolate DNA ranging from 250 to 550 bp in size. Ligation fragments marked by biotin were immobilized using MyOne Streptavidin C1 DynaBeads (Invitrogen) and ligated to paired-end adapters (Illumina). Hi-C libraries were then amplified using PE PCR 1.0 and PE PCR 2.0 primers (Illumina) with eight PCR amplification cycles.

## Biotinylated RNA bait library design
Biotinylated 120-mer RNA baits were designed as previously described[22]. Briefly, to target both ends of HindIII restriction fragments that overlap Ensembl promoters of protein-coding, noncoding, antisense, snRNA, miRNA, and snoRNA transcripts. A target sequence was valid if its GC content ranged between 25 and 65% and the sequence contained no more than two consecutive Ns and was within 330 bp of the HindIII restriction fragment terminus. The full bait list is supplied in the source data file.

## Promoter capture Hi-C
Capture Hi-C of promoters was carried out with SureSelect target enrichment, using the custom-designed biotinylated RNA bait library and custom paired-end blockers according to the manufacturer's instructions (Agilent Technologies). After library enrichment, a post-capture PCR amplification step was carried out using PE PCR 1.0 and PE PCR 2.0 primers with four PCR amplification cycles. CHi-C libraries were sequenced on the Illumina HiSeq 2000 platform for paired-end sequencing.

## Promoter capture Hi-C analysis
Paired-end fastq files were processed through the HiCUP pipeline[38]. Briefly, Hi-C junctions are identified between read-pairs, and each side of the interaction are mapped to their HindIII digested regions on the hg19 genome, duplicate and artefactual reads are then filtered out.

Significant interactions were called on both individual and merged replicate HiCUP bam files using CHiCAGO[39]. Briefly, CHiCAGO calls interactions by creating a background model based on random 'Brownian collisions' between chromatin fragments and technical components within the data and computes $p$-values based on the expected true positive rates. $P$-values are adjusted taking into consideration interaction distances (lower numbers of interactions are expected over longer distances). Interaction scores are then computed by $-\log$ transforming these adjusted $p$-values. Any interaction with a final CHiCAGO score $>= 5$ was considered as a significant interaction. Across CHi-C libraries, interactions were considered replicated or overlapping if they fell within 4 kb of the interacting fragment.

Differential CHi-C interactions were called using the statistical framework of GOTHiC[40]. Here, each replicate pair (PML::RARA induced vs Control) was directly compared. For each interaction, the probability of observing a given number of read pairs by chance is modeled where the expected values are the number of reads observed in the control condition. For multiple testing correction, we used Independent Hypothesis Weighting (IHW v.1.6.0) with an FDR cut off $<= 0.01$. A consensus differential interaction set was created by combining interactions that were present in at least two replicates.

## Omni ATAC-seq
ATAC-seq libraries for zinc-induced and non-induced libraries were generated following the protocol outlined previously[41]. Briefly, $5 \times 10^5$ cells were re-suspended in ATAC-Resuspension buffer (0.1% NP40, 0.1% Tween-20, and 0.01% Digitonin). Isolated nuclei were then incubated in a transposition mixture containing 100 nM final transposase at 37 °C for 30 min. DNA was extracted and purified using the Zymo DNA Clean and Concentrator-5 Kit. Libraries were pre-amplified with five cycles using NEBNext Master Mix, additional cycles were determined by SYBR Green qPCR. DNA was purified (Zymo) after amplification and quantitated using the NEBNext quant kit by qPCR. Paired-end libraries were sequenced on the Illumina HiSeq 2000 platform.

## Omni ATAC-seq analysis
Reads were trimmed of adapters and low-quality reads using Trimmomatic[33] and aligned to the hg19 genome using Bowtie2. PCR duplicates were removed using Samtools and blacklisted reads were removed. Peaks were called using Macs2 peak caller, with a $q$-value threshold of 0.01.

For the differential ATAC analysis, DiffBind (Deseq2) was used. A merged set of all peaks from both replicates were used as bins to call peaks. Any bin with an FDR adjusted $p$-value $<= 0.1$ was considered as a significant differential ATAC-seq peak.

## Cut&Run nuclei isolation
After five hours of incubation, induced and non-induced U937-PR9 cells were washed in fresh RPMI/(10% FBS) medium. Cells were then treated with a hypertonic solution and mild detergent (IGEPAL) followed by clarification through a sucrose gradient to isolate nuclei. Cut&Run was then performed on isolated nuclei as described[19]. In brief, nuclei were re-suspended in Cut&Run wash buffer and incubated with 4 µg antibody (PML::RARA, ab43152, Abcam; RARA, C15310155, Diagenode) for 12 h at 4 °C. Nuclei were then washed in Cut&Run wash buffer and incubated with PA-MNase for two hours at 4 °C. Nuclei were washed again in Cut&Run wash buffer, then incubated with $CaCl_2$ for 20 minutes at 0 °C to activate the PA-MNase. Stop buffer containing 20 mM EDTA was added to cells to stop the $CaCl_2$ induced PA-MNase activity. Nuclei were then incubated at 37 °C with RNase, allowing PA-MNase cut DNA fragments to diffuse into the supernatant. DNA was precipitated using phenol: chloroform/ethanol. Libraries were then prepared using the NEBNext DNA Library prep master mix set and multiplex oligos for Illumina (New England Biolabs). Library quality was assessed using Tapestation E220 high sensitivity (Agilent), and qPCR quantitation was performed using NEB library quant kit. Libraries were sequenced using an Illumina HiSeq 2500 instrument for single-end sequencing.

## Cut&Tag
Cut&Tag was performed on NB4 cells using the Active Motif Cut&Tag-IT kit, based on ref. [20], using the same PML::RARA antibody as used for Cut&Run (ab43152). Libraries were sequenced on an Illumina MiSeq instrument for paired-end sequencing.

## Cut&Run/Tag analysis
Reads were trimmed of adapters and low-quality reads using Trimmomatic[33] and aligned to the hg19 genome using Bowtie2. PCR duplicates were removed using Samtools and blacklisted reads were removed using Bedtools. Peaks were called using the SEACR peak calling algorithm, which is designed for accurate peak calling of Cut&Run experiments. Paired control experiments were used as background and the 'norm union' settings were used as thresholding. We identified reproducible peaks across replicates using the Irreproducible Discovery Rate (IDR) according to ENCODE guidelines with an IDR < 0.05 threshold. All experiments were carried out in duplicate.

## RARA, PML, and H3K9/K14ac U937-PR9 ChIP-seq
Raw fastq files (2 x RARA, 4 x PML, and 4 x H3K9/K14ac) were downloaded from GEO accession GSE18886[15]. Reads were trimmed of adapters and low-quality reads using Trimmomatic[33] and aligned to the

hg19 genome using Bowtie2. PCR duplicates were removed using Samtools, blacklisted reads were removed, and peaks were called using Macs2 peak caller using the paired uninduced experiment as background, with a q-value threshold of 0.01. The 3315 consensus RARA/PML peaks used for overlapping the Cut&Run data was created by taking any RARA peak that overlapped with one of the two PML dataset peaks. We identified reproducible peaks across replicates using the Irreproducible Discovery Rate (IDR) according to ENCODE guidelines with an IDR < 0.05 threshold.

### RARA ChIP-chip
Processed regions identified by Wang et al.[12] were downloaded and lifted over to the hg19 genome using the "BSgenome.Hsapiens.UCSC.hg19" and "import chain" R packages.

### Functional enrichment
Gene Ontology analysis of up/downregulated genes was performed using EnrichR and ontologies were taken from the GO Biological Process database. Pathway analysis was performed on differentially expressed genes using PathFindR. Here we used the adjusted p-value and fold changes associated with each gene as weights for pathway enrichment within the Kegg database.

### Genome annotation
For the annotation/assignment of genomic features to datasets, we used the TxDb.Hsapiens.UCSC.hg19.knownGene database. Here a feature was assigned to a data point if the apex of the peak (Cut&Run or ATAC-seq) overlapped with the given feature.

### Motif enrichment
For the enrichment of motifs at ATAC-seq peaks associated with the six categorical outcomes, we used the Homer motif suite of tools. Homer uses a differential motif enrichment algorithm, so for each enrichment calculation, we used a background of sampled ATAC-seq peaks from a mix of all the other peaks in the categories not being compared. We used parameters 'findMotifsGenome.pl' with '-size −200,200' to center peaks to a 400 bp region and '-bg' to set our background of peaks.

### Data preparation for XGBoost
For each ATAC-seq peak in each of the six categories, the motif binding site repertoire was identified using Homer 'annotatePeaks.pl' using the '-size −200,200' parameter. Here we counted the number of binding sites on each fragment for each of the 436 motifs in the Homer database, centered around a 400 bp region. This gave us a 436 x n fragments matrix which we applied column-wise log10 transformation.

For 'one vs one' pairwise comparisons, dataset sizes were randomly balanced towards the smaller dataset.

For 'one vs all' comparisons, the 'all' dataset was created from subsampling an equal number of interactions from all other categories, keeping the comparison balanced.

### Data preparation for the APL patient vs CD34 + XGBoost models
A consensus ATAC-seq peak set was created by merging all Macs2 called peaks from two in-house APL patients and two publicly available GEO CD34 + datasets (GSE96772). Each raw sequencing file was processed according to the ATAC-seq methods section.

Differentially expressed genes were generated by comparing the two APL patient sample RNA-seq and two CD34 + RNA-seq files from GEO (GSE96772). The same Limma framework was used as described in the RNA-seq methods section. Genes with an FDR adjusted p-value <0.01 were considered different.

Differential interactions (DI's) were generated using GOTHiC comparing each APL patient to each publicly available CD34 + CHiC dataset (Array express: E-MTAB-10701). Consistent DI's were considered if present in at least two comparisons and in the same direction.

ATAC-seq peaks were then assigned to each of the six expressional categories: gained/upregulated in CD34 + cells; gained/no change in CD34 + cells; gained/downregulated in CD34 + cells; gained/downregulated in APL patients; gained/no change in APL patients; gained/upregulated in APL patients. Peaks were gained assigned a motif repertoire using HOMER—as described above.

### XGBoost
Hyperparameter tuning for optimal model performance was performed using the MLR package, focusing on 'min child weight' and 'max depth'. A grid search was applied to cycle through 400 unique combinations of the 2 hyperparameters keeping other parameters fixed. Constant parameters were as follows: subsample = 0.8, colsample by tree = 0.8, eta = 0.1, and gamma = 1. The training was performed on 80% of the data and tested on 20% of the data, with five-fold cross-validation. Optimal parameters giving the best AUC scores after the grid search were then used to train and test on five unique subsets of each category to encompass 100% of the data. This gave, for each of the six categories, five model outputs which were used for explorative analysis by SHAPELY.

### SHAPELY analysis
For each of the six categories and their five models, data points were kept if the model predicted correctly. SHAPELY was then applied to the data matrix, giving a weighted contribution of predictive features for each motif. SHAPELY weighting were assigned to clusters using DBSCAN[42].

### Statistics and reproducibility
Using the PROPER R package we determined that with two replicates we have 80% power to detect differential gene expression for those genes that are expressed at a level of at least 10 reads per library. For RNA-seq, we used the variance derived from replicates to identify reproducible differences between conditions. For Cut&Run and ATAC-seq experiments we used the irreproducible discovery rate (IDR) to identify reproducible signals. The number of reproducible peaks determined that two replicates were sufficient[43]. RNAseq, Cut&Run, Cut&Tag, and ATAC-seq libraries were carried out with two biological replicates. For the capture Hi-C, the cell line libraries were performed with three biological replicates and the patient sample libraries were performed in duplicate (two different patients). In capture Hi-C experiments we only kept differential interactions present in both replicates. We used five-fold cross validation for machine learning models. No data were excluded from analysis. Experiments were not randomized. The investigators were not blinded to allocation during experiments and outcome assessment. The statistical tests used for each figure are supplied in Supplementary Table 1.

### Reporting summary
Further information on research design is available in the Nature Portfolio Reporting Summary linked to this article.

## Data availability
Raw sequencing data for RNA-seq, Cut&Run, Cut&Tag, ATAC-seq, and promoter capture Hi-C for the cell lines and ex vivo patient samples have been deposited in the GEO database under the series GSE173755. Publicly available datasets used in this study are: GSE18886 (U937-PR9 ChIP-seq), GSE96772 (CD34 + RNA-seq and ATAC-seq), GSE137662 (U937-PR9 time course RNA-seq) and E-MTAB-10701 (CD34 + Capture Hi-C). Source data are provided with this paper and publicly available at: https://doi.org/10.5281/zenodo.7467566.

## Code availability

The custom code associated with this study is publicly available at https://github.com/borimifsud/REBEL.

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

## Acknowledgements

We thank S. Henikoff for the generous gift of pA-MNase, used in the CUT&RUN assays. This study was supported by an MRC doctoral studentship (MR/N013700/1, W.V.) and additional funding from Blood Cancer UK (14007, C.S.O.; 15023, R.D.), NIHR (RP-PG-0108-10048, R.D.), and CRUK (A29806, R.D.). Additional support for next-generation sequencing was provided by the National Institute for Health Research (NIHR) Biomedical Research Centre based at Guy's and St Thomas' NHS Foundation Trust and King's College London. The project was enabled through access to the MRC eMedLab Medical Bioinformatics infrastructure, award MR/L016311/1.

## Author contributions

W.V., R.D., and C.S.O. conceived the study. W.V., B.M., R.D., and C.S.O. designed the experiments. W.V. conducted the RNA-seq experiments. W.V. conducted the Cut&Run experiments with participation from A.K. and P.L. X.H. carried out the ATAC-seq experiments. W.V. and C.S.O. conducted the capture Hi-C experiments. W.V. carried out the machine learning experiments with input from A.E. and H.B. W.V., B.M., and C.S.O. analyzed and interpreted the data. J.K.C. and W.V. conducted the Western blotting and Cut&Tag experiments. W.V., B.M., and C.S.O. prepared the manuscript. C.S.O. supervised the work and directed the project.

## Competing interests

The authors declare no competing interests.
