## [Peer Review file · Nature Communications]

Reviewers' comments:

Reviewer #1 (Remarks to the Author): Expert in Acute Promyelocytic Leukemia clinical research and genomics

Villiers et al. used integrated multi-omics and reported context-specific gene regulatory activities of PML-RARA in APL. They analyzed RNA-seq, cut&run, ATAC-seq, and Hi-C datasets, mostly in an inducible cell line model (U937-PR9). This study showed some technical advances. However, this work showed little conceptual advance, and most conclusions were similar to some work previously published. The paper is predominantly a genomics paper and lacks molecular and biological studies to confirm the findings.

Major comments:

1. This study was mainly performed on the U937-PR9 cell model, which expresses huge amount of PML-RARA and is difficult to finely tune the level of PML-RARA expression. This cell model has been associated to a number of artefacts in past studies and could clearly lead to binding to low affinity DNA sites that are not recognized in real APL cells.
2. The finding of context-specific gene regulatory activities of PML-RARA in APL is not novel. Tan et al. have reported the dual function of PML-RARA in APL using the multi-omics approach in NB4 cells (PMID: 32854112). They have knocked down the PML-RARA in NB4 cells and identified the dual function of PML-RARA in transcriptional regulation in APL, which is similar to the conclusion in this study.
3. The concept of the engagement of PML-RARA in chromatin conformation regulation has also been reported recently by Wang et al (PMID: 32393309). They have also applied the U937-PR9 model and investigated the impact of PML-RARA on chromatin interaction using the CHIA-PET approach. They reported the repression function of PML-RARA in chromatin conformation regulation.
4. Another major issue is that there are a number of published studies which are directly relevant to this manuscript, but the authors do not properly give credit to these other authors for their contributions.
5. Most results were descriptive data analysis. The author should provide evidence on what factors determines the transcriptional repression or transcriptional activation activity of PML-RARA; what determines the enhanced or decreased chromatin interaction during PML-RARA induction; whether is related to the post-translational modification of PML-RARA; and what is the interaction with well-known co-factors (such as RXR)?

Minor comments:

There are a host of minor issues with the manuscript, but when you add them all up they detract from the overall quality.

1. The mRNA and protein levels of PML/RARa expression in U937 and U937-PR9 cells should be shown.

2. Examples used in Figure 3, 4, 6 and supplemental figures were different. To obtain robust conclusions, the author might show the cut&run, ATAC-seq, Hi-C results in same genes or regions.

3. The cut&run appear to be a very useful approach to identified binding sites of PML/RARa on chromatin. However, the cells were unfixed in this approach, and the nuclei were treated with antibodies for 12 hours, this might introduce some artificial binding of PML/RARa and some newly identified PML/RARa binding sites might be artificial as well. The author should validate some of these newly identified targets by ChIP-seq, ChIP-qPCR, or any other experimental evidence.

4. For better visualization, differential promoter-enhancer interactions should be calculated by HOMER or other tools and showed as heatmaps.

Reviewer #2 (Remarks to the Author): Expert in Acute Promyelocytic Leukemia and PML-RARA fusion

In this report, the authors have explored an inducible model of PML/RARA expression to explore, in a very extensive manner, the genomic changes resulting for expression of this oncoprotein. This involves RNA-seq, ATAC-seq, cut and run and capture Hi-C. As agreed with the editor, this review will not concentrate on the genomics analyses by themselves, but on the importance of these studies for our understanding of the APL model and pathogenesis.

First, the cellular model, while technically convenient, has a number of serious caveats. It is an immortalized cell-line, the promoter leaks so that PML/RARA expression can be detected at low levels before Zn administration and the level of expression post-Zn induction is much higher than in APL cells, precluding physiological relevance. The system is also polluted by the very strong Zn-initiated stress response clearly visible on Fig. 1B.

Second, multiple previous studies have reported on this issue and it is unclear what is conceptually new in this report, except that more technologies were used. At large, all published genomics studies have demonstrated that PML/RARA is an altered transcription factor, but failed to identify critical determinants of pathogenesis. The Wang study, in Blood, already reported that PML/RARA can be an activator of a significant number of genes. In that sense, the abstract is somehow misleading when stating that this issue is "under-explored". It is actually unclear to this reviewer what are the novel conclusions drawn from this study?

Third, it is unclear to what extent many of the loose correlations or overlaps shown between gene sets or binding sites, only reflect similar differentiation status.

Reviewer #3 (Remarks to the Author): Expert on Hi-C and PCHi-C techniques

Comments on Villiers et al

In this paper, Villiers et al. examine PML-RARA binding in a model cell line, and the relationship with chromatin interactions and RNA-Seq. They also examine patient samples. They use a machine learning model to try to identify combinatorial motifs of transcription factors that might predict PML-RARA transcriptional responses. Overall the paper is well written and the figures are well prepared. The multi-omics data is performed well, but I am concerned about the way that the machine learning was performed (see detailed comments below) and the way the paper has been written, there are a lot of findings that have been presented without much discussion, and I am also unclear as to how these findings make a difference in our understanding of Acute Promyelocytic Leukemia or PML-RARA treatment.

1. APL is one of the most treatable forms of leukemia, so I am not sure about the significance of this work. Is there anything we can learn from this work that will help us to better help patients with other subtypes of acute myeloid leukemia, which have a much more dismal 5-year response rate?
2. What was the rationale behind the choice of 5 hours after zinc sulphate induction for performing the assays?
3. Given that there are many regions where there is PML-RARA binding but no change in gene expression (Figure 2C-D), what could be the functional consequences of this binding? Or is there no functional consequence?
4. "Over 97% of PML-RARA peaks were localized centrally at pre-existing ATAC-seq peaks, suggesting that the fusion protein is recruited to sites where other factors are already bound" If HDAC inhibitors or

HAT inhibitors or some other drugs are used prior to PML-RARA activation in order to reduce the pre-existing ATAC-Seq peaks, does this prevent fusion protein recruitment?

5. PML-RARA patients have been greatly helped by drugs such as arsenic trioxide and all-trans retinoic acid (ATRA). I was wondering if these results could also help to explain the good results seen with arsenic trioxide and ATRA? How do chromatin interactions and RARA binding change upon arsenic trioxide or ATRA treatment?

6. I am very concerned that a high percentage of the dataset was used for machine learning (models were first trained by 80% of the data) and the lack of cross-validation or other validation models. In previous machine learning methods such as TargetFinder, after the paper was published there were criticisms of the work (e.g. Xi and Beer, PLoS Computational Biology, 2018; Cao and Fullwood, Nat Genet., 2019 which found that the results may have been inflated because of the use of the random sampling of the data which led to data being used both in training and testing datasets). I think there needs to be more validation here, both in computationally as well as wet lab (e.g. knockdown of two or more transcription factors that have shown to interact to see if this leads to changes in the chromatin interactions).

7. Can the machine models be used to make predictions with the patient data?

8. Why would the PML-RARA bound genes show more divergence in the patient data? What could be the possible reasons? Is it because there were only 2 patients examined and just these few might not be fully representative?

9. "We also identified genes that had discordant gene expression levels across patients and cell line yet exhibited highly similar interaction profiles (Figure S6L)" => What could be the possible reasons?

Reviewer #4 (Remarks to the Author): Expert in multi-omics analysis and machine learning methods

The fusion of Promyelocytic Leukemia (PML) and Retinoic Acid Receptor Alpha (RARA) gene is the primary cause of Acute Promyelocytic Leukemia (APL). However, the mechanisms of how PML-RARA interacts and regulates downstream gene expression are rarely reported. In this study, the authors carried out an experiment for RNA-seq, ATAC-seq, CUT&RUN, and Hi-C for identifying the regulatory patterns of PML-RARA. They claimed a novel machine learning method to be used in finding local transcriptomic factor binding sites, revealing distinct signatures that modulate PML-RARA transcriptional responses.

Major comments

1. It is not clear how many replicates/samples were included for each of the sequencing analyses. A solid power analysis is a must to convince the audience of all the statistical analyses performed in this study.

2. The authors included 10 public CHIP-seq datasets and combined them with their own CUT&RUN data, and more public data should also be included for other data types to increase the sample pool.
3. Direct analysis on individual data types may include biases for association studies. There are a lot of tools for the purpose of bulk data integration to remove such bias, which should be considered.
4. The author claimed a novel machine learning method to deconvolute but simply using XGBoost in an existing package. It is definitely not an innovation. What are negative and positive training groups? A justification should be given for the proper choice of machine learning tool, as even for XGBoost, there are many enhanced versions. The justification of power to train the model should be given. Also, it is recommended to do cross-validation using at least one more method.
5. There is no clue where the patient samples are collected. Minimum information about the two patient samples was given, making it hard to evaluate the results claimed related to pattern comparison between cell line and patients. Also, patient samples always contain much higher heterogeneities, both sample-width and cellular width. Single-cell level data analysis should be applied for more accurate representations.
6. There is research published in 2018 claiming a distinctive epigenome signature in APL using multi-omics data. The authors should cite this paper and discuss the difference between the two works.

Villiers et al. Multi-omics and machine learning reveal context-specific gene regulatory activities of PML-RARA in Acute Promyelocytic Leukemia

We thank each Reviewer for their valuable expertise and comments. Here, we respond to each comment.

Reviewer #1 (Remarks to the Author): Expert in Acute Promyelocytic Leukemia clinical research and genomics

Villiers et al. used integrated multi-omics and reported context-specific gene regulatory activities of PML-RARA in APL. They analyzed RNA-seq, cut&run, ATAC-seq, and Hi-C datasets, mostly in an inducible cell line model (U937-PR9). This study showed some technical advances. However, this work showed little conceptual advance, and most conclusions were similar to some work previously published. The paper is predominantly a genomics paper and lacks molecular and biological studies to confirm the findings.

Major comments:

1. This study was mainly performed on the U937-PR9 cell model, which expresses huge amount of PML-RARA and is difficult to finely tune the level of PML-RARA expression. This cell model has been associated to a number of artefacts in past studies and could clearly lead to binding to low affinity DNA sites that are not recognized in real APL cells.

The reviewer implies that the bulk of the additional binding sites that we detect, compared to previous studies (Martens et al., 2010; Wang et al., 2010) is a mere consequence of higher PML-RARA expression in the induced U937-PR9 cells, rather than a considerably more sensitive detection assay (Cut&Run). While the U937-PR9 cells have been reported to express high levels of PML-RARA upon induction, we find through quantification of the fusion transcript from the RNA-seq data that its expression is only slightly more than that which is detected in NB4 cells, another APL cell line model, which harbours a t(15;17 PML-RARA) translocation. In addition to the cell line RNA-seq libraries, we have also included quantification of RNA-seq libraries from nine APL patient samples, including the two samples that are analysed in our study. We note that there is in fact a high degree of variability in fusion gene expression levels across the APL patient cohort. Our analyses show that fusion transcript levels in the U937-PR9 and NB4 cell lines are broadly similar to some patient samples (Rebuttal Figure 1A). This illustrates that the cell lines are suitable models for PML-RARA expression levels for at least some patients. We now include this analysis in Supplementary Figure 1.

We attest that the high number of PML-RARA binding sites we have detected is due to the sensitivity of the Cut&Run assay - not dependent on extreme expression levels of the fusion in the U937-PR9 cell line system. To support this, we have now applied Cut&Tag, a variant of the Cut&Run assay to NB4 cells using the PML-RARA fusion antibody to compare it with previous ChIP-seq profiling. Here we detect 10,963 PML-RARA peaks, encompassing 68% of known sites from previous ChIP-seq data plus an additional 9,753 peaks (Rebuttal Figures 1B-C). We attempted several times to obtain complementary PML-RARA Cut&Run datasets from cryopreserved APL patient samples but were unsuccessful due to the fragility of these cells in these assays. Regardless, our evidence suggests that the Cut&Run/Tag technique accounts for the high number of novel peaks we detect, compared to ChIP-seq methods that have been employed in previous studies.

Moreover, our conclusions about the expression outcomes of PML-RARA binding are not affected by the additional peaks. For instance, our data indicate that PML-RARA recruitment to 80% of the

binding sites does not significantly alter the expression of the associated gene. This observation does not change if we only examine the most significant binding sites; assessment of the top 3,315 PML-RARA binding sites identified in the U937-PR9 Cut&Run assays (equal to the number of PML-RARA binding sites identified by ChIP-seq) shows that 76% of the associated genes do not significantly alter their expression levels upon PML-RARA recruitment (Rebuttal Figure 2). Similarly, the proportions of up- and downregulated PML-RARA bound genes remain consistent.

2. The finding of context-specific gene regulatory activities of PML-RARA in APL is not novel. Tan et al. have reported the dual function of PML-RARA in APL using the multi-omics approach in NB4 cells (PMID: 32854112). They have knocked down the PML-RARA in NB4 cells and identified the dual function of PML-RARA in transcriptional regulation in APL, which is similar to the conclusion in this study.

Yes, Tan and colleagues published in 2021 their observation that PML-RARA is also bound to the promoters of genes that are upregulated. This itself was not the first report; Wang et al. (J Cell Biochem) showed in 2018 that MYB is upregulated. We cite both papers and make no claims that ours is a novel observation. However, neither paper observes that the vast majority (80%) of PML-RARA binding genes do not show any transcriptional response. The Tan paper is not without its shortcomings. Firstly, they have employed a knockdown approach to reduce PML-RARA levels within the NB4 cell line. Knockdowns are not 100% efficient and low levels of transcript will remain. Secondly, not all the data provided by Tan et al. was supported by replicated datasets (e.g. ChIP-seq data). This precludes any rigorous analysis.

Regardless, we contend that the main novelty of our manuscript lies in our work to discern and characterise the different types of PML-RARA binding environments, presented later in the paper.

3. The concept of the engagement of PML-RARA in chromatin conformation regulation has also been reported recently by Wang et al (PMID: 32393309). They have also applied the U937-PR9 model and investigated the impact of PML-RARA on chromatin interaction using the CHIA-PET approach. They reported the repression function of PML-RARA in chromatin conformation regulation.

Again, we do not contend that we are the first to publish on the long-range interactions in this model system and we do cite this paper. Wang et al. (2020) was also published during our study. They did not report any long-range interaction in the context of either upregulated or unchanged gene expression genes. Moreover, the Wang paper was published without replicated datasets for any of their libraries, and we are very surprised that it passed a rigorous review process.

4. Another major issue is that there are a number of published studies which are directly relevant to this manuscript, but the authors do not properly give credit to these other authors for their contributions.

We have not knowingly omitted any references, so any that are missing reflects an oversight. At the same time, we would expect the reviewer to provide at least some justification for this criticism and provide at least a few examples.

5. Most results were descriptive data analysis. The author should provide evidence on what factors determines the transcriptional repression or transcriptional activation activity of PML-RARA; what determines the enhanced or decreased chromatin interaction during PML-RARA induction; whether is related to the post-translational modification of PML-RARA; and what is the interaction with well-known co-factors (such as RXR)?

While post-translational modification of PML-RARA may also contribute to the differential transcriptional output of PML-RARA bound genes, our study has focussed on the cooperating binding sites that modulate PML-RARA activity. The identification of the transcription factors that bind to these sites represents a major undertaking that extends well beyond the scope of this manuscript and would significantly increase its size beyond that of a standard submission. Moreover, it would detract from the central message of the paper that demonstrates the application of machine learning to distil complex transcription factor binding site patterns.

Minor comments:

There are a host of minor issues with the manuscript, but when you add them all up they detract from the overall quality.

1. The mRNA and protein levels of PML/RARa expression in U937 and U937-PR9 cells should be shown.

As discussed in Point 1, we now include the quantitation of PML-RARA fusion transcripts from the U937-PR9 cells and across a wider cohort of patient RNA-seq libraries (n=9) to demonstrate the modest levels of expression in the cell line (Rebuttal Figure 1A, Supplementary Figure 1A).

2. Examples used in Figure 3, 4, 6 and supplemental figures were different. To obtain robust conclusions, the author might show the cut&run, ATAC-seq, Hi-C results in same genes or regions.

We chose examples that best illustrated the observations made on a genome-wide scale. Our conclusions are robust because they are supported by rigorous statistical analyses. We note that profiles for PTGER4 are shown in Figures 3 and 6. For the Reviewer's benefit, we show here its ATAC-seq profile alongside each other assay (Rebuttal Figure 3), reflective of Figure 4. ATAC-seq peaks around the PTGER4 gene do not change drastically, therefore it is not an illustrative example for Figure 4 in the manuscript.

3. The cut&run appear to be a very useful approach to identified binding sites of PML/RARa on chromatin. However, the cells were unfixed in this approach, and the nuclei were treated with antibodies for 12 hours, this might introduce some artificial binding of PML/RARa and some newly identified PML/RARa binding sites might be artificial as well. The author should validate some of these newly identified targets by ChIP-seq, ChIP-qPCR, or any other experimental evidence.

The Cut&Run method and its derivative (Cut&Tag) have been used in hundreds of published studies since its original publication in 2017 (Skene and Henikoff, eLife). It largely has replaced the use of ChIP-seq methods due to its simplicity, extremely low background, low cell number requirement and tolerance of antibodies that are incompatible with fixation. We disagree with the reviewer about the exclusion of fixation contributing to artificial binding and false positive binding sites. Indeed, fixation is by far more prone to noisy signal. This is also seen in comparisons between fixed and native ChIP libraries. In any case, we already show comparisons of our Cut&Run derived data and the ChIP-seq data from Martens et al. and Wang et al. (Figure S2A). It clearly demonstrates that we detect most peaks identified by others. Furthermore, our analyses displayed further on in Figure S2 show the strength and specificity of the peaks that we identify. We can call more peaks because the background in these assays is very low. This contrasts with the Martens and Wang reports that were hindered both by high backgrounds and the need to overlap signals for PML and RARA antibodies, since the antibody against the PML-RARA fusion was unavailable to them. It is unclear whether the PML-RARA antibody would work well in a ChIP-seq experiment as it has not been validated for this assay. We do show however that it works very well in Cut&Run.

4. For better visualization, differential promoter-enhancer interactions should be calculated by HOMER or other tools and showed as heatmaps.

The use of Homer is inappropriate for calling long-range interactions in capture Hi-C experiments since it does not handle properly the enriched (captured) side of the di-tag (See Osborne and Mifsud, 2017 Chromosome Research (<https://link.springer.com/content/pdf/10.1007/s10577-016-9546-4.pdf>). Instead, we have used well-established interaction calling pipelines (GOTHic and CHiCAGO) that are tailored for these assays. We do not agree with the reviewer with the requirement for heatmaps. We have applied stringent statistical cut-offs to identify true interacting fragments. This information is muddled through the display of heatmaps.

Reviewer #2 (Remarks to the Author): Expert in Acute Promyelocytic Leukemia and PML-RARA fusion

In this report, the authors have explored an inducible model of PML/RARA expression to explore, in a very extensive manner, the genomic changes resulting for expression of this oncoprotein. This involves RNA-seq, ATAC-seq, cut and run and capture Hi-C. As agreed with the editor, this review will not concentrate on the genomics analyses by themselves, but on the importance of these studies for our understanding of the APL model and pathogenesis.

First, the cellular model, while technically convenient, has a number of serious caveats. It is an immortalized cell-line, the promoter leaks so that PML/RARA expression can be detected at low levels before Zn administration and the level of expression post-Zn induction is much higher than in APL cells, precluding physiological relevance. The system is also polluted by the very strong Zn-initiated stress response clearly visible on Fig. 1B.

The notion that the promoter in U937-PR9 cells 'leaks' expression originates (to the best of our knowledge) from the Martens paper, who noted that some peaks found at sites in induced cells were also present in uninduced cells. They specifically identified the UNC458 gene as one such example. We contend that the apparent leakiness is an artefact of their experimental ChIP-seq system, which suffered from high off-target signal and potentially false positive signals. Examination of the UNC458 locus in our datasets displays no such peaks in the uninduced cells, however there is a strong peak in the induced cells (Rebuttal Figure 4). Moreover, quantifying the expression levels of PML-RARA in the U937-PR9 RNA-seq data detected zero fusion transcripts in the uninduced state (Rebuttal Figure 1A). We conclude that at the very least, there are minimal levels of PML-RARA transcript prior to zinc induction.

While the U937-PR9 cells have been reported to express high levels of PML-RARA upon induction, we find through quantification of the fusion transcript from the RNA-seq data that its expression is only slightly more than that which is detected in NB4 cells, another APL cell line model, which harbours a t(15;17 PML-RARA) translocation. In addition to the cell line RNA-seq libraries, we have also included quantification of RNA-seq libraries from nine APL patient samples, including the two samples that are analysed in our study. We note that there is in fact a high degree of variability in fusion gene expression levels across the APL patient cohort. Our analyses show that fusion transcript levels in the U937-PR9 and NB4 cell lines are broadly similar to some patient samples (Rebuttal Figure 1A). We now include this analysis in Supplementary Figure 1A.

It is true that we observe a strong upregulation of genes that are responsive to zinc, which we acknowledge in the manuscript. However, we disagree that this amounts to 'pollution' that would skew the results across the genome. Indeed, functional enrichment analysis shown in Figure 1 of the manuscript demonstrates that the genes that are most impacted transcriptionally are those involved in myeloid differentiation and cancer. We corroborated our RNA-seq data with patient samples and showed a high degree of concordance for differentially-expressed genes.

Second, multiple previous studies have reported on this issue and it is unclear what is conceptually new in this report, except that more technologies were used. At large, all published genomics studies have demonstrated that PML/RARA is an altered transcription factor, but failed to identify critical determinants of pathogenesis. The Wang study, in Blood, already reported that PML/RARA can be an activator of a significant number of genes. In that sense, the abstract is somehow misleading when stating that this issue is "under-explored". It is actually unclear to this reviewer what are the novel conclusions drawn from this study?

We feel that by the reviewer's own admission, they were more focussed on the significance to APL than the genomics and perhaps has not fully appreciated the analyses carried out to discern and characterise the different types of PML-RARA binding environments that channel distinctive signalling pathways to invoke specific and varied outcomes. These findings plus the machine learning applications we have applied are novel and significant. We stand by our statement in the abstract indicating that the impact on gene regulation and expression is under-explored. No one else has combined transcription, long-range interactions, fusion gene recruitment and chromatin occupancy to fully examine the outcome of binding, including the 80% of promoter recruitments that do not have a transcription change impact. This observation has been overlooked by every other published study.

Third, it is unclear to what extent many of the loose correlations or overlaps shown between gene sets or binding sites, only reflect similar differentiation status.

What is clear is that genes grouped within clusters by virtue of their transcription factor binding motif environments are statistically enriched for distinct ontologies. If there is a similar differentiation response amongst these genes, it is likely driven through these shared regulatory inputs.

Reviewer #3 (Remarks to the Author): Expert on Hi-C and PCHi-C techniques

Comments on Villiers et al

In this paper, Villiers et al. examine PML-RARA binding in a model cell line, and the relationship with chromatin interactions and RNA-Seq. They also examine patient samples. They use a machine learning model to try to identify combinatorial motifs of transcription factors that might predict PML-RARA transcriptional responses. Overall the paper is well written and the figures are well prepared. The multi-omics data is performed well, but I am concerned about the way that the machine learning was performed (see detailed comments below) and the way the paper has been written, there are a lot of findings that have been presented without much discussion, and I am also unclear as to how these findings make a difference in our understanding of Acute Promyelocytic Leukemia or PML-RARA treatment.

1. APL is one of the most treatable forms of leukemia, so I am not sure about the significance of this work. Is there anything we can learn from this work that will help us to better help patients

with other subtypes of acute myeloid leukemia, which have a much more dismal 5-year response rate?

Our study was not born from a motivation to directly develop therapies for leukaemia patients but instead to use a tractable model to better understand the early regulatory alterations that can follow soon after cell exposure to a potent oncogenic transcription factor. APL provides this and is supported by a detailed molecular understanding of the cellular defect that can be treated through targeted therapies.

Having said this, APL treatment is not without its challenges. While representing a great success story for targeted molecular therapies, approximately 10% of patients are classified as high-risk and prone to relapse that is refractory to further treatment.

2. What was the rationale behind the choice of 5 hours after zinc sulphate induction for performing the assays?

Our choice of five-hour stimulation was based upon conditions set by Martens et al., 2010 Cancer Cell, who in turn based their conditions on the paper that originally described the cell line (Grignani et al., 1993 Cell). By using the same conditions, we were able to benchmark our datasets against theirs.

3. Given that there are many regions where there is PML-RARA binding but no change in gene expression (Figure 2C-D), what could be the functional consequences of this binding? Or is there no functional consequence?

We believe that this may be a significant observation. It is possible that there is no functional consequence of PML-RARA binding at these sites, or perhaps they require further activation signals to take effect. Clearly, PML-RARA doesn't normally 'belong' in the nucleus, and it is possible that its recruitment is incompatible with initiating transcriptional effects at certain sites. It would be of interest, albeit beyond the scope of this project, to determine how often non-oncogenic transcription factors also get recruited to sites without impact.

4. "Over 97% of PML-RARA peaks were localized centrally at pre-existing ATAC-seq peaks, suggesting that the fusion protein is recruited to sites where other factors are already bound" If HDAC inhibitors or HAT inhibitors or some other drugs are used prior to PML-RARA activation in order to reduce the pre-existing ATAC-Seq peaks, does this prevent fusion protein recruitment?

The recruitment of PML-RARA to pre-existing ATAC-seq sites is an auxiliary observation. We believe it is not of direct relevance to the manuscript and that these experiments, while interesting, would not add significantly to our understanding of how PML-RARA function is modulated by its transcription factor binding environment.

5. PML-RARA patients have been greatly helped by drugs such as arsenic trioxide and all-trans retinoic acid (ATRA). I was wondering if these results could also help to explain the good results seen with arsenic trioxide and ATRA? How do chromatin interactions and RARA binding change upon arsenic trioxide or ATRA treatment?

The molecular mechanisms by which ATRA and As₂O₃ specifically target the PML-RARA fusion protein are well studied. ATRA increases the physiological concentrations of RA to overcome the dominant interactions of cofactors to PML-RARA. This allows RA to bind directly to the RARA moiety and

PML-RARA subsequently dissociates from DNA and is degraded. As2O3 also triggers the degradation of PML-RARA, in this case the mechanism is thought to involve direct binding of the PML moiety leading to its SUMOylation and subsequent proteasomal degradation. We would anticipate that exposure to ATRA and As2O3 would revert the long-range interaction patterns to the patterns that existed prior to PML-RARA expression. While we agree that this may be interesting, we believe that it is beyond the scope of this study to examine how PML-RARA action is altered by the local transcription factor binding environment.

6. I am very concerned that a high percentage of the dataset was used for machine learning (models were first trained by 80% of the data) and the lack of cross-validation or other validation models. In previous machine learning methods such as TargetFinder, after the paper was published there were criticisms of the work (e.g. Xi and Beer, PLoS Computational Biology, 2018; Cao and Fullwood, Nat Genet., 2019 which found that the results may have been inflated because of the use of the random sampling of the data which led to data being used both in training and testing datasets). I think there needs to be more validation here, both in computationally as well as wet lab (e.g. knockdown of two or more transcription factors that have shown to interact to see if this leads to changes in the chromatin interactions).

We did not use random sampling of the same data to calculate the error rate of misclassification. For each 'one vs all' model, we separated the fragments based on their associated functional outcomes (e.g. gained interactions/upregulated (GU); lost interaction/downregulated (LD)). We then added an equal number of fragments randomly selected from the other functional outcome groups. For example, 1,201 fragments were categorised as GU were paired with 1,201 non-GU fragments. These datasets were used for training and testing. To ensure there was no inadvertent bias in our paired datapoint subsets, we repeated the training/testing one hundred times using a different, randomly selected paired dataset and found that the AUC scores were stable (± 0.02 std dev) (Rebuttal Figure 5). Moreover, five-fold cross validation was performed for all iterations of each model, again with stable results. It suggests that subsampling of the negative class does not impact the performance of the model.

We ensured for each model there was zero overlap of fragments within the test and train sets. Unlike the TargetFinder methodology, which had the caveat that an interaction pair could have one partner shared with another data point in another class, thus it could be present in both test and train – we treat each interaction fragment end as a unique data point, and ensured no fragments were able to be present in multiple classes prior to running our models. We note that the AUCs varied from model to model. The best 'one vs all' model achieved 0.7 AUC, which is a modest score; if there was 'leakage' between categories, we would expect greater scores.

We used the XGBoost machine learning framework to generate our models and have now benchmarked it against an independent machine learning framework, LightGBM. Using identical sampling techniques and five-fold cross-validation, we obtain comparable AUCs to the XGBoost models. Interrogation of the LightGBM generated models using the SHAP analyses as described in the manuscript also identified multiple unique subclusters for each category, including those with similar TFBS repertoires to the XGBoost models (see Rebuttal Figure 5). Taken together, it demonstrates our observations are robust and not an artefact of the specific machine learning algorithm that we employed.

Regarding the wet lab experimentation for validation, we point out that we are characterising transcription factor binding sites, rather than the TFs themselves.

7. Can the machine models be used to make predictions with the patient data?

In principle these models could be applied to the patient data if they had differential comparisons made to appropriate healthy control data, such as from CD34+ cells. We also envisage that these machine learning methodologies and data formats can be applied to other datasets where differential information and transcription factor binding site data are available.

8. Why would the PML-RARA bound genes show more divergence in the patient data? What could be the possible reasons? Is it because there were only 2 patients examined and just these few might not be fully representative?

The PML-RARA bound subset of genes are those that show the largest difference upon induction, therefore they are likely to exhibit higher divergence, as well. Furthermore, the differences might be due to the different genetic background in the patient samples, as additional mutations might also contribute to the observed differences.

9. “We also identified genes that had discordant gene expression levels across patients and cell line yet exhibited highly similar interaction profiles (Figure S6L)” => What could be the possible reasons?

We didn't want to speculate too much in the manuscript. This represented a very small proportion of the genes. It could for instance reflect a change in factors that are mediating the long-range interactions. For example, activating bound factors could replace repressing bound factors.

Reviewer #4 (Remarks to the Author): Expert in multi-omics analysis and machine learning methods

The fusion of Promyelocytic Leukemia (PML) and Retinoic Acid Receptor Alpha (RARA) gene is the primary cause of Acute Promyelocytic Leukemia (APL). However, the mechanisms of how PML-RARA interacts and regulates downstream gene expression are rarely reported. In this study, the authors carried out an experiment for RNA-seq, ATAC-seq, CUT&RUN, and Hi-C for identifying the regulatory patterns of PML-RARA. They claimed a novel machine learning method to be used in finding local transcriptomic factor binding sites, revealing distinct signatures that modulate PML-RARA transcriptional responses.

Major comments

1. It is not clear how many replicates/samples were included for each of the sequencing analyses. A solid power analysis is a must to convince the audience of all the statistical analyses performed in this study.

Data on replicates were presented in the supplementary material and replicates were combined according to best practices in order to use only high confidence observations in downstream analyses. RNAseq, Cut & Run and ATACseq libraries were carried out in duplicate. For the capture Hi-C, the cell line libraries were performed in triplicate and the patient sample libraries were performed in duplicate. For clarity, we now include this information in the methods section of the manuscript.

2. The authors included 10 public ChIP-seq datasets and combined them with their own CUT&RUN data, and more public data should also be included for other data types to increase the sample pool.

We did not combine the Cut&Run data, which profiles PML-RARA fusion binding across the genome, with the PML and RARA ChIP-seq profiles. Instead, we used these libraries as a benchmark. The other publicly available ChIP-seq data profiles we used were to map genome-wide epigenetic marks. To our knowledge there are no other publicly available ATAC-seq nor capture Hi-C datasets. We have used available U937-PR9 RNA-seq data for our analysis of expression levels over time, shown in Figure S2R. It is not appropriate to merge these with our own, due to the differences in stimulation time.

3. Direct analysis on individual data types may include biases for association studies. There are a lot of tools for the purpose of bulk data integration to remove such bias, which should be considered.

We are uncertain what the reviewer means by association studies. We would appreciate it if the reviewer could point us in the direction of such tools.

4. The author claimed a novel machine learning method to deconvolute but simply using XGBoost in an existing package. It is definitely not an innovation. What are negative and positive training groups? A justification should be given for the proper choice of machine learning tool, as even for XGBoost, there are many enhanced versions. The justification of power to train the model should be given. Also, it is recommended to do cross-validation using at least one more method.

We do not consider it incorrect to state that we have used a new machine learning strategy to deconvolute the complex regulatory input of transcription factors, because this strategy has not previously been reported in addressing this type of question. The algorithm is not new but the application in this way is. We have clarified this by changing the text to: ‘we develop a novel, computational machine learning application to deconvolute the complex, local transcription factor binding site environment’.

In each model, the positive group consists of ATAC-seq containing fragments belonging to one of the regulatory/transcription outcome categories (e.g., gained interaction and upregulated) versus either another single category or a balanced mixture of all other categories. For each ‘one vs all’ model, we separated the fragments based on their associated functional outcomes (e.g. gained interactions/upregulated (GU); lost interaction/downregulated (LD)). We then added an equal number of fragments randomly selected from the other functional outcome groups. For example, 1,201 fragments were categorised as GU were paired with 1,201 non-GU fragments. These datasets were used for training and testing. To ensure there was no inadvertent bias in our paired datapoint subsets, we repeated the training/testing one hundred times using a different, randomly selected paired dataset and found that the AUC scores were stable (± 0.02 std dev) (Rebuttal Figure 5). Moreover, five-fold cross validation was performed for all iterations for each model, again with stable results. It suggests that subsampling of the negative class does not impact the performance of the model.

The XGBoost algorithm was chosen based on its successful application in winning Kaggle AI competitions (<https://www.kaggle.com/competitions>), plus its adept handling of highly complex datasets. In our hands, XGBoost did perform well to differentiate between the outcome categories. We have now validated

our findings using an independent, enhanced gradient boosting framework –lightGBM. Using identical sampling techniques and five-fold cross-validation, we obtain comparable AUCs to the XGBoost models. Interrogation of the LightGBM generated models using the SHAP analyses as described in the manuscript also identified multiple unique subclusters for each category, including those with similar TFBS repertoires to the XGBoost models (see Rebuttal Figure 5). Taken together, it demonstrates our observations are robust and not an artefact of the specific machine learning algorithm that we employed.

5. There is no clue where the patient samples are collected. Minimum information about the two patient samples was given, making it hard to evaluate the results claimed related to pattern comparison between cell line and patients. Also, patient samples always contain much higher heterogeneities, both sample-width and cellular width. Single-cell level data analysis should be applied for more accurate representations.

Information on the patient samples was inadvertently omitted from the manuscript and is now included. Patient samples were collected from the Haematology Unit, Guy's Hospital, London, UK. Each sample contained >90% blast cells and were used for RNA extraction and capture Hi-C library preparation. Samples of fresh peripheral blood were donated with informed consent from patients at diagnosis. (IRAS project ID: 220344, REX reference 06/Q0702/140). These details have been added to the methods section of the manuscript.

6. There is research published in 2018 claiming a distinctive epigenome signature in APL using multi-omics data. The authors should cite this paper and discuss the difference between the two works.

We believe that the reviewer is referring to the publication by Singh et al. (Oncotarget). It profiles epigenetic signatures (H3 acetylation, H3K27me3 and DNA methylation) specifically for high-risk APL versus low-risk patients. It does not interrogate the context of PML-RARA recruitment and function, as our study achieves. We do not compare patients in different risk categories in our study. The patient samples were low risk only, therefore we cannot corroborate the findings of the Singh paper.

Rebuttal Figure 1

A) Barplots showing the normalised number of PML-RARA fusion transcripts detected using the Arriba Fusion detection tool. Briefly each raw RNA-seq dataset (U937-PR9 induced/uninduced, NB4, and 9 in-house APL patients) was used as input into the Arriba tool (<https://genome.cshlp.org/content/31/3/448>) and the number of reads that span both PML and RARA exons were counted, these were normalised by the number of total reads for that sample.

B) Venn diagram showing the overlap of the 10,963 peaks from Cut&Tag in the NB4 cells and the 1,787 peaks from the ChIP-seq data from Martens et al 2010.

C) Peak tracks of 4 genic regions showing the U937-PR9 PML-RARA Cut&Run peaks (blue) and the NB4 PML-RARA Cut&Tag peaks (red) at 4 differentially expressed genes referenced in the main paper.

Rebuttal Figure 2

A

Differential Expression of ALL PML-RARA (promoter) bound genes

B

Differential Expression of the top PML-RARA (promoter) bound genes - using the top 3315 peaks (based on U937-PR9 PR ChIP-seq data having 3315 peaks)

Rebuttal Figure 2

A) The same figure as the main paper (Figure 2C-D) showing a venn diagram overlapping the PML-RARA bound genes and the up- and downregulated genes. The significance of each overlap is represented as the p-value using a chi-squared test. The ranked plot shows the expression level distribution of all 6,242 PML-RARA-bound genes in the PML-RARA-induced U937-PR9 cells, as compared to uninduced expression. Genes are ordered according to their expression level in uninduced cells. The expression of each gene in the uninduced cells is represented by the central orange line. The PML-RARA-induced expression of each gene is indicated by green, grey and red dots indicating upregulation, no expressional change and downregulation respectively. Expressional change is based on genes having an adjusted p-value <=0.05. The side bar plot shows the proportion of PML-RARA-bound genes that are upregulated, have no expressional change or downregulated.

B) The same plots as Figure 2A however only using the top (strongest) 3,315 PML-RARA peaks and their associated gene promoters.

Rebuttal Figure 3

Rebuttal Figure 3 - Reviewer 1

This is the same as Figure 3I in the paper, however the ATAC-seq track for the induced (blue) and uninduced (orange) U937-PR9 cells has been added.

A

Rebuttal Figure 4

A) Genomic tracks of PML-RARA binding profiles at the UNC458 gene locus. Normalised PML-RARA read density in uninduced (orange) and induced (blue) U937-PR9 cells are shown. Blue indicates the PML-RARA induced profile and gold indicates the PML-RARA uninduced (background) profile. Y-axis represents the read count normalised by library size. The genomic coordinates are indicated along the x-axis. Gene bodies and exons are indicated by connecting blue blocks.

Rebuttal Figure 5

Rebuttal Figure 5

A) Bar plot showing the AUC scores for each one-vs-all machine learning model using XGBoost.

Dashed line (0.5) represents random predictions. The error bars show the standard deviation from the mean of 100 iterations of negative class subsampling experiments. AUC = Area Under the Curve.

B) Identical plot to A, however using LightGBM instead of XGboost.

C) Multiple plots similar to the paper figure 5D-F. The plot is split by a dotted line for downstream analysis of XGBoost (left) and lightGBM (right) model predictions. The first tSNE plot visualises the clustering of ATAC-seq peaks based on the SHAPELY Interpretation scores derived from each machine learning model. Each dot is colored according to the interaction/expression category to which the fragment containing the ATAC-seq peak belongs. The second tSNE is identical to the first however the data is clustered and each cluster is assigned a unique colour. The final plot on each side shows the Protein-protein Interaction network of TFs, which bind the top 15 predictive motifs in a representative gained-upregulated cluster from each model's output. Circles are TFs and lines show known and predicted physical interactions between them as identified by the STRING database. PML and RARA are denoted as stars.

REVIEWER COMMENTS

Reviewer #1 (Remarks to the Author):

The authors have not really addressed my concerns about the conceptual advances of their findings over earlier works and the functional or experimental evidence of their conclusions that can be drawn at this stage. Particularly, the expression level of PML-RARa (Rebuttal Figure 1A) seems incorrect in U937-PR9 cells including both before and after induction. The authors ignored the assays on the protein levels. The validation of PML-RARa binding sites should be compared between PR9 and NB4 cells (or patient samples). Current validation between chip-seq and CUT-and-Tag in NB4 cells does not support the authenticity of PML-RARa binding sites. Also, the author did not provide the analyzed data with Supplemental Tables as well as the raw data (only RNA-seq data under the GEO accession GSE173754, the token for cut and run data is not available). More importantly, authors claimed the sensitivity of Cut-and-Run which helped them find that the fusion protein was recruited to sites where other factors were already bound, but they denied the importance of identifying the factors which influence the PML-RARa binding and determine the transcriptional outcome. The same representative example genes were essential, at least the examples in Cut-and-run should include those in ATAC-seq and Hi-C. At the current stage, these issues are sufficiently important as to preclude publication of this study.

The authors claim that they include the cut-and-run data under the GEO accession GSE173754. However, only RNA-seq data was included (see attached below).

Series GSE173754	
Status	Private until May 01, 2023 Private data, not to be shared or distributed without permission
Title	Multi-omics and deep learning reveal context-specific gene regulatory activities of PML-RARA in Acute Promyelocytic Leukemia [RNA-seq]
Organism	Homo sapiens
Experiment type	Expression profiling by high throughput sequencing
Summary	The PML-RARA fusion protein is the hallmark driver of Acute Promyelocytic Leukemia (APL) and disrupts retinoic acid signaling, leading to wide-scale gene expression changes and uncontrolled proliferation of myeloid precursor cells. While known to be recruited to binding sites across the genome, its impact on gene regulation and expression is under-explored. Using integrated multi-omics datasets, we characterize the influence of PML-RARA binding on gene expression and regulation in an inducible cell line model and APL patient ex vivo samples. We find that genes whose regulatory elements recruit PML-RARA are not uniformly transcriptionally repressed, as commonly suggested, but also may be upregulated or remain unchanged. We apply a novel computational deep learning strategy to deconvolute the complex, local transcription factor binding site environment at PML-RARA bound positions to reveal distinct signatures that modulate how PML-RARA directs the transcriptional response.
Overall design	Application of RNA-seq to the U937-PR9 cell line system, before and after PML-RARA inunction. Application of RNA-seq to two ex vivo APL patient samples
Contributor(s)	Villiers W , Dillon R , Mifsud B , Osborne C
Citation missing	Has this study been published? Please login to update or notify GEO. Note that private accession will be released, in accordance to guidelines.
Submission date	May 03, 2021
Last update date	Apr 29, 2022
Contact name	Cameron Osborne
E-mail(s)	cameron.osborne@kcl.ac.uk
Organization name	King's College London
Department	Medical and Molecular Genetics
Street address	Guy's Hospital Campus
City	London
ZIP/Postal code	SE1 9RT
Country	United Kingdom
Platforms (1)	GPL16791 Illumina HiSeq 2500 (Homo sapiens)
Samples (6) Less...	GSM5277837 U937-PR9 Control rep1 RNA-seq GSM5277838 U937-PR9 Control rep2 RNA-seq GSM5277839 U937-PR9 Induced rep1 RNA-seq GSM5277840 U937-PR9 Induced rep2 RNA-seq GSM5277841 APL patient 1 RNA-seq GSM5277842 APL patient 2 RNA-seq
Relations	
BioProject	PRJNA726969
SRA	SRP318229
External data have been provided but are not accessible for review while status is private.	

Reviewer #2 (Remarks to the Author):

I am unconvinced by many of the points put forward in response to the comments of reviewers 1 and 2. The U937 system has been around for almost 30 years and many groups have used it. All reported much higher levels of PML/RARA expression by Western blot, when compared to APL patients or NB4. The mRNA assessment by the authors is likely not reflecting protein levels.

Similar to reviewer 1, my major comment remains : what do we actually learn from these studies? It is textbook knowledge, for example for nuclear receptors, that transcription binding does not necessarily translate into transcriptional control. That an altered transcription factor, with a high affinity for DNA (because of the PML-mediated dimerization of the RARA/RXRA complex) and reduced binding site specificity engages into different types of chromatin interactions is hardly surprising. The dual effect (activation or repression) of PML/RARA on some of its targets was the main message of a very detailed previous publication in Blood.

It is not because the authors acknowledge the overwhelming Zn response, that this not does blur the data.

local transcription factor binding site environment at PML-RARA bound positions to reveal distinct signatures that modulate how PML-RARA directs the transcriptional response (abstract)...

What is the actual evidence for that? The machine learning signatures? The authors describe them as very complex, suggesting that they do not have much biological relevance ? It is the only novelty put forwards in the abstract.

Comments of APL clinical aspects are not accurate. The key remaining issue in clinical trials are the (rare) early deaths.

The authors do not discuss the possibility that many of the long-range interactions are a mere reflection of the changing differentiation profile.

Finally, in the revised manuscript, modifications are still apparent in the supplementary data, but changes made in the main text are not highlighted, which does not facilitate their identification.

Reviewer #3 (Remarks to the Author):

My concerns have been addressed.

Reviewer #4 (Remarks to the Author):

The authors have fixed some of my concerns, however, there are still several issues left.

1. For question 3, as the authors kept referring multi-omics data integration (e.g., RNA-seq integrate with CUT& RUN, ATAC-seq, and Hi-C), I expected to see details about how data are integrated. Also, the authors should justify why existing tools for omics data integration, such as mentioned in <https://doi.org/10.1039/D0MO00041H>, are not suitable for their study.
2. For question 4, I would like to see more details about the model training. For example, what parameters have been optimized and how the default parameters were determined.
3. For question 5, the authors did not respond to my concerns regarding the heterogeneity difference between cell line data and patient data. Also, why single-cell data was not considered.

Reviewer #5 (Remarks to the Author): Expert in CUT&RUN, CUT&Tag, and ATAC-seq

This revised manuscript has already been reviewed by four referees, each with different areas of expertise. I am largely confining my review to the evaluation of genomic profiling data (mainly CUT&RUN and ATAC-seq, along with CUT&Tag data provided in the rebuttal).

The CUT&RUN data appear to be of high quality. Although replicates have different numbers of peaks in some conditions, they overlap reasonably well (Fig. S2B), suggesting the difference in peak numbers may reflect a thresholding artifact that includes some low-level peaks in one replicate that are missed in another. The authors indicate that they use peaks that overlap in both replicates for downstream comparisons/analyses, which is appropriate. In the rebuttal, the authors included CUT&Tag data from NB4 cells that anecdotally appear to overlap with their U937-PR9 induced CUT&RUN data at four loci.

However, to better address Reviewer 1's concern about these data, the authors should compare the overlap of their CUT&RUN and CUT&Tag peaks throughout the genome, rather than comparison of the CUT&Tag data with previous ChIP data, and it is worth including the CUT&Tag data in the supplement even if just for QC. Nonetheless, I agree with the authors that the increased signal to noise of CUT&RUN likely accounts for the 'extra' peaks relative to ChIP-seq.

In addition, I agree with the authors that the lack of fixation in the CUT&RUN protocol is a positive, allowing fixation/shearing mediated artifacts to be eliminated. In addition, crosslinking has been shown to have minimal effects on CUT&RUN. Finally, the lack of peaks in the uninduced sample suggests the effect of 'leaky' PML-RARA expression is negligible. However, if this concern remains, *it should be trivial to measure uninduced and induced protein levels (not mRNA) of PML-RARA by Western Blotting to put this issue to rest.*

The discussion of the ATAC-seq results seems to deemphasize an important finding. The authors write "While not entirely correlated, genes with PML-RARA bound promoters and differential ATAC-seq peaks were more likely to be downregulated than upregulated (Figure S4F)." However, the largest group of genes with differential ATAC peaks (essentially all of which show reduction in ATAC signal) are genes that do not change in expression. Although chromatin accessibility is thought to be one of the best markers of regulatory element activity, here, reduction of accessibility has no effect on gene expression in a large majority of cases. This may suggest that while accessibility of a regulatory element at 'steady state' (in unperturbed cells) may be a marker of gene activity, reduction of accessibility during a perturbation (induction of PML-RARA in this case) does not necessarily reveal a reduction in gene activity. *The authors may want to discuss this point further.*

Finally, although I am not an expert in APL, I do note there is a benefit to a study that integrates expression, binding, accessibility and 3D folding data to draw conclusions from the integrated dataset, even if one or more of each of these approaches has been previously performed in PML-RARA models. Furthermore, although it is anecdotally understood that many TF binding sites are ineffectual for activating or repressing transcription, there are to my knowledge relatively few studies where this is addressed systematically, which is also useful to the genomics and gene expression communities.

Two minor comments:

1. There are numerous figure panels in the main figures and supplement in which the text labels (including but not limited to the labels of axes) are so small they are nearly unreadable. Some examples include Figs. 1c, 3i, 3j, 4b, 5f, S2b, and S5a.
2. Line 113: "We applied the Cut&Run method to identify PML-RARA binding sites due to its low signal-to-noise ratio and..." I think you mean high signal-to noise (relative to ChIP-seq).

Reviewer #1 (Remarks to the Author)

The authors have not really addressed my concerns about the conceptual advances of their findings over earlier works and the functional or experimental evidence of their conclusions that can be drawn at this stage.

In addition to observations that corroborate the findings of others (PML-RARA binding elements engage in long-range interactions, PML-RARA repress some genes and activates others), we find that the vast majority of PML-RARA recruitment does not invoke a transcriptional response (novel finding). Our investigations use machine learning (novel methodological application) to dissect the transcription factor binding site environments that modulate PML-RARA activity. We identify distinct signatures that map not only the transcriptional impact but also the impact on long-range interactions (novel finding). We observe multiple distinct transcription factor binding site configurations for each functional outcome (novel finding). We note that many of these interactions are present in ex vivo samples derived from patients with APL (novel finding).

Particularly, the expression level of PML-RARa (Rebuttal Figure 1A) seems incorrect in U937-PR9 cells including both before and after induction.

Our expression level analyses of the PML-RARA transcript are correct. In these analyses, we have used the Arriba fusion detection tool to quantitate the amount of PML-RARA transcript within the RNA-seq libraries. The presented bar plot displays the read number of fusion transcripts (determined based on the paired-end Illumina sequencing) relative to the total number of reads in that sample. This approach enables the fusion transcripts to be isolated from the non-rearranged PML and RARA alleles.

The authors ignored the assays on the protein levels.

As requested, we now provide Western blot analysis of PML-RARA protein levels in uninduced and induced U937-PR9 cell line, NB4 cell line, MV4-11 cell line (which does not harbour a PML-RARA rearrangement) and three primary samples from APL patients (Figure S1B in the paper, Rebuttal Figure 1A-B). Our blot shows that the induced U937-PR9 cells have the highest amounts of PML-RARA protein. However, PML-RARA expression in patient samples is not massively different, ranging from 40% to 70% of the expression in the induced U937-PR cell line. In contrast, PML-RARA expression in NB4 cells is considerably lower, ranging from 20 to 40% of patient levels. The expression levels in the patient samples are variable meaning that there is no standardised protein amount among patients. It shows that the U937-PR9 cell line is a suitable model for PML-RARA expression for at least some patients. Moreover, our analyses of PML-RARA recruitment in NB4 cells and a patient sample, described below, show that PML-RARA protein levels does not influence its recruitment to the new binding sites we have identified.

We note that while there is little detectable PML-RARA protein in MV4-11 cells, there is potentially some low-level detection in uninduced U937-PR9 cells. To our eyes, the bands do not exactly match to the PML-RARA bands in other lanes, which may be indicative of non-specific bands, although this is open to interpretation. The presence of PML-RARA in uninduced cell samples would be incongruent with both our RNA-seq data, where no fusion transcripts were detectable and our Cut&Run data, which demonstrated virtually no PML-RARA peaks. It is conceivable that what little PML-RARA protein is present, its binding is negligible.

The validation of PML-RARa binding sites should be compared between PR9 and NB4 cells (or patient samples).

In response to the first round of reviews, we measured the PML-RARA binding sites in NB4 cells using CUT&Tag. We presented the overlap of signals in induced U937-PR9 and NB4 cells at four loci, which demonstrated a strong correlation. We now also include the total overlap of binding sites genome-wide. It

shows that 67% of the binding sites detected in NB4 cells are also present in induced U937-PR9 cells (Rebuttal Figure 2A). We now include this analysis in the manuscript (Figure S2L).

Additionally, we have also carried out CUT&Tag analysis for PML-RARA binding in a primary sample derived from an APL patient, patient #3 in the Western blot. It demonstrates that 15,225 PML-RARA peaks are detectable in the patient sample, of which 5,829 (38%) were also present in the U937-PR9 cell line, representing a significant overlap ($X^2 = 9.7^{e-22}$). This was also consistent with the overlapping of the NB4 and the APL patient sample (Rebuttal Figure 2B-C).

The signal-to-noise ratio was considerably lower in the patient sample, possibly due to inherent cellular fragility of the primary sample, which hindered detection of many peaks (Rebuttal Figure 2D). However, irreproducibility discovery rate (IDR) calculation demonstrated that 8,119 peaks had consistent patterns between the patient and the cell line datasets, yielding an overall IDR score of 2.6x, which is indicative of very strong reproducibility between these heterologous libraries (Rebuttal Figure 2E-F). It demonstrates high concordance of PML-RARA binding sites between the induced U937-PR9 cell line and the primary APL patient sample.

Next, we interrogated the 5,829 PML-RARA binding peaks that were consistent between the patient and U937-PR9 samples. We note that collectively the subset of peaks was significantly stronger than the subset that was not consistent (Rebuttal Figure 3A). The 5,829 peaks overlapped with 2,738 genes, including key up- and downregulated genes such as *CEBPA* and *SP1* (Rebuttal Figure 3B-C). The expression direction of the 2,738 genes upon induction of the U937-PR9 cells (7% upregulated, 17% downregulated and 76% no expression change) showed the consistent subset display a highly similar distribution to what we report in the manuscript for all PML-RARA U937-PR9 peaks (8% upregulated/12% down regulated/80% no expression change) (Rebuttal Figure 3D). The genomic distribution of the consistent peaks was also in line with previous analysis with 50% of peaks at promoters, 22% intergenic and 25% intronic (Rebuttal Figure 3E).

Collectively, these analyses show that the binding patterns observed in the U937-PR9 cell line are consistent with patterns in NB4 cells and a primary APL patient. The number of peaks detected in each sample is not a consequence of protein expression level. The transcriptional observations, including binding and no expressional change, hold true to putative sites observed in an APL patient sample.

Current validation between chip-seq and CUT-and-Tag in NB4 cells does not support the authenticity of PML-RARa binding sites.

The point of these experiments is to demonstrate that the CUT&Tag/CUT&RUN methodologies detect considerably more PML-RARA peaks than ChIP-seq. There is a wealth of published studies that support the fact that CUT&RUN and CUT&Tag are more sensitive than ChIP-seq. For example, please refer to Skene and Henikoff, *eLife* 2017; Kaya-Okur et al., *Nature Communications*, 2019.

Also, the author did not provide the analyzed data with Supplemental Tables as well as the raw data (only RNA-seq data under the GEO accession GSE173754, the token for cut and run data is not available).

We apologise for this oversight as all datasets were intended to be made available. While all data was uploaded under GEO accession GSE173754, the token was generated inadvertently for only the RNA-seq. This problem has been rectified.

More importantly, authors claimed the sensitivity of Cut-and-Run which helped them find that the fusion protein was recruited to sites where other factors were already bound, but they denied the importance of identifying the factors which influence the PML-RARa binding and determine the transcriptional outcome.

Firstly, all PML-RARA recruitment is to sites where other factors are already bound. This is true for the binding sites that have been identified by others using ChIP-seq as well as the additional sites that we detect.

Our data presented in Figure 4A demonstrates that virtually no new ATAC-seq peaks are formed upon PML-RARA induction.

The identification of the transcription factor binding environment at these PML-RARA recruitment sites that may modulate its activity is precisely the point of the second half of the paper. As we have stipulated before, we focus on the transcription factor binding sites rather than the putative transcription factors themselves. We point out that multiple TFs are capable of binding to a given binding site and speculatively chasing specific factors that might be recruited would quickly descend into a time-consuming and expensive endeavour. Our machine-learning-led characterisations of the PML-RARA binding sites show that the binding environment composition and syntax is highly complex. It simply is not within the scope of this study to carry out tens of ChIP-seq/CUT&RUN assays to try to assemble the factors involved at these sites.

The same representative example genes were essential, at least the examples in Cut-and-run should include those in ATAC-seq and Hi-C.

We provided the reviewer with an example of a gene, PTGER4 displaying ATAC-seq, CUT&RUN and differential capture Hi-C tracks. We don't agree that this adds anything substantial to the manuscript. Indeed, we assert that it makes the figures unnecessarily busy. As we point out in our earlier response to the reviewers, our conclusions are robust because they are supported by rigorous statistical analyses.

At the current stage, these issues are sufficiently important as to preclude publication of this study.

Reviewer #2 (Remarks to the Author)

I am unconvinced by many of the points put forward in response to the comments of reviewers 1 and 2. The U937 system has been around for almost 30 years and many groups have used it. All reported much higher levels of PML/RARA expression by Western blot, when compared to APL patients or NB4. The mRNA assessment by the authors is likely not reflecting protein levels.

As requested, we now provide Western blot analysis of PML-RARA protein levels in uninduced and induced U937-PR9 cell line, NB4 cell line, MV4-11 cell line (which does not harbour a PML-RARA rearrangement) and three primary samples from APL patients (Figure S1B in the paper, Rebuttal Figure 1A-B). Our blot shows that the induced U937-PR9 cells have the highest amounts of PML-RARA protein. However, PML-RARA expression in patient samples is not massively different, ranging from 40% to 70% of the expression in the induced U937-PR cell line. In contrast, PML-RARA expression in NB4 cells is considerably lower, ranging from 20 to 40% of patient levels. We highlight again that expression levels in the patient samples are variable meaning that there is no standardised protein amount among patients. It shows that the U937-PR cell line is a suitable model for PML-RARA expression for at least some patients. Furthermore, our newly included CUT&Tag analyses of PML-RARA binding in NB4 cells, which exhibited the lowest protein level of the samples tested by Western blot, plus the patient sample data shown in this rebuttal demonstrates that protein amount has little impact on PML-RARA recruitment. There is very strong overlap of binding sites between these samples and induced U937-PR9 cells.

We note that while there is little detectable PML-RARA protein in MV4-11 cells, there is potentially some low-level detection in uninduced U937-PR9 cells. To our eyes, the bands do not exactly match to the PML-RARA bands in other lanes, which may be indicative of non-specific bands, although this is open to interpretation. The presence of PML-RARA in uninduced cell samples would be incongruent with both our RNA-seq data, where no fusion transcripts were detectable and our Cut&Run data, which demonstrated virtually no PML-RARA peaks. It is conceivable that what little PML-RARA protein is present, its binding is negligible.

Similar to reviewer 1, my major comment remains: what do we actually learn from these studies? It is textbook knowledge, for example for nuclear receptors, that transcription binding does not necessarily translate into transcriptional control. That an altered transcription factor, with a high affinity for DNA (because of the PML-mediated dimerization of the RARA/RXRA complex) and reduced binding site specificity engages into different types of chromatin interactions is hardly surprising. The dual effect (activation or repression) of PML/RARA on some of its targets was the main message of a very detailed previous publication in Blood.

We do not agree that this transcription factor binding without a transcriptional effect is ‘textbook knowledge’ and are unaware of any reports where this has been characterised in any detail. As described by Reviewer 5, this phenomenon, while perhaps anecdotally observed, has not previously been studied systematically, using complementary measures of transcription, as we present in this manuscript. We find it remarkable that such an overwhelming proportion (80%) of the recruitment events do not seem to manifest a transcriptional response.

The reviewer questions what is learnt from these studies. We find that the vast majority of PML-RARA recruitment does not invoke a transcriptional response (novel finding). Our investigations use machine learning (novel methodological application) to dissect the transcription factor binding site environments that modulate PML-RARA activity. We identify distinct signatures that map not only to the transcriptional impact but also the impact on long-range interactions (novel finding). We observe multiple distinct transcription factor binding site configurations for each functional outcome (novel finding). We note that many of these interactions are present in ex vivo samples derived from patients with APL (novel finding).

It is not because the authors acknowledge the overwhelming Zn response, that this not does blur the data.

All cell lines have limitations to their applicability and the U937-PR9 cell line requires exposure to zinc to express PML-RARA. We cannot deduce what secondary impact zinc exposure has on PML-RARA recruitment. However, what is clear is that there is a very strong correlation between the induced cell line and patient-derived samples for transcription (measured by RNA-seq) and long-range promoter interactions (measured by promoter capture Hi-C). Moreover, we now provide a PML-RARA binding profile (measured by CUT&Tag) for an APL patient and NB4 cells, which demonstrates a strong overlap of binding profiles (Rebuttal Figures 1 and 2). Collectively, it shows that expression, long-range interactions and PML-RARA recruitment are not reliant on exposure to zinc.

local transcription factor binding site environment at PML-RARA bound positions to reveal distinct signatures that modulate how PML-RARA directs the transcriptional response (abstract)...

What is the actual evidence for that? The machine learning signatures? The authors describe them as very complex, suggesting that they do not have much biological relevance? It is the only novelty put forwards in the abstract.

The evidence for this is that the model correctly predicts the behaviours of the different elements based upon their transcription factor binding site compositions. The signatures are distinct between the functional classes and between subsets within classes. Yes, the number of features used by the machine learning algorithm to correctly predict the functional outcome of the element is large, implying that the signature is complex. However, there is no suggestion that the signature has little biological relevance. Indeed, the fact that the prediction is dependent upon so many modulating elements that contribute incrementally is in itself a significant observation.

Comments of APL clinical aspects are not accurate. The key remaining issue in clinical trials are the (rare) early deaths.

We agree that a critical remaining issue in APL is early death (ED) within 30 days of diagnosis from bleeding events. This is more common than reported in clinical trials, for example in population-based analysis ED rates are as high as 29% in Sweden (Lehmann, et al., 2011 *Leukemia*), 17.3% in the United States (Park, et al.,

2011 *Blood*) and 21.8% in Canada (Paulson, et al., 2014 *BJH*). The majority (>60%) of ED is attributed to coagulopathy and haemorrhagic events (Mantha et al, 2016 *Blood*). In addition to this risk of early death from haemorrhage, patients entering complete remission do remain at risk of relapse. Those diagnosed with HR disease, by virtue of a high white blood cell count, have a 30% chance of relapse following the standard ATRA/anthracycline based therapy (Sanz, et al., 2000 *Blood*).

The authors do not discuss the possibility that many of the long-range interactions are a mere reflection of the changing differentiation profile.

Firstly, uninduced U937-PR9 cells are not differentiating. Secondly, it is well established that the expression of PML-RARA creates a differentiation block, so it's highly unlikely that it will alter the non-differentiating status upon induction. Thirdly, the cells are induced for only five hours before assaying. In such a short period, none of the cells will have undergone a full cell cycle.

Additionally, any alterations to long-range interactions that occur through differentiation reflect the rewiring of promoter contacts to distal regulatory elements that shape the transcriptional response, rather than a generic side effect.

Finally, in the revised manuscript, modifications are still apparent in the supplementary data, but changes made in the main text are not highlighted, which does not facilitate their identification.

We have endeavoured to highlight all modifications to the text in the revised version.

Reviewer #3 (Remarks to the Author)

My concerns have been addressed.

Reviewer #4 (Remarks to the Author)

The authors have fixed some of my concerns, however, there are still several issues left.

1. For question 3, as the authors kept referring multi-omics data integration (e.g., RNA-seq integrate with CUT& RUN, ATAC-seq, and Hi-C), I expected to see details about how data are integrated. Also, the authors should justify why existing tools for omics data integration, such as mentioned in <https://doi.org/10.1039/D0MO00041H>, are not suitable for their study.

We have performed a multi-stage analysis for data integration. As pointed out by the paper highlighted by the reviewer (Graw et al. 2021), decisions on the type of 'omics data integration need to be taken in view of the biological questions asked. In our manuscript, we aimed to classify interacting fragments into distinct categories based on multiple omics datasets (RNA-seq, capture Hi-C, ATAC-seq, CUT&RUN). To achieve this, for each dataset type, the data is classified (RNA-seq: no change/upregulated/downregulated; capture Hi-C: gained interactions/lost interactions; ATAC-seq: gained peaks/lost peaks/no change peaks; CUT&RUN: PML-RARA binding/not binding). For each element that is engaged in a differential interaction (both the promoter and its distal interacting partner), the characteristics are collated (e.g., upregulated/gained interaction/no change ATAC peak/PML-RARA binding). No special tools are available to perform these analyses, therefore after analyses of the individual datasets, data were integrated using R. We are aware that the choice of thresholds for individual datasets can influence such analysis, however, we observed little variation when alternative thresholds were used. Furthermore, the machine learning algorithm was able to distinguish the categories created based on the TF motifs found in them, which implies that the categories represent biologically meaningful classes.

The paper by Graw et al., which was published at a very late stage in our project, lists publicly available resources of datasets that can be integrated into studies. Where appropriate, we have included such data in

our analyses. This paper also lists software that can be used to assist dataset integration from a variety of experimental sources. While tools exist to aid the integration of transcriptomics and ChIP-seq/Cut&Run dataset types, there were no tools described and we are not familiar with any tool that is specifically designed to also accommodate Hi-C/capture Hi-C libraries.

2. For question 4, I would like to see more details about the model training. For example, what parameters have been optimized and how the default parameters were determined.

For the model optimisation within the manuscript, we focused on optimising two hyperparameters: the max depth and min child weight. We used a grid search with 20 different values for each parameter which cycled through all 400 unique combinations of the 2 hyperparameters keeping other parameters fixed. For constant parameters the default XGBoost settings were used as follows: subsample = 0.8, colsample by tree = 0.8, eta = 0.1 and gamma = 1. The parameters from the model with the best AUC after the grid search were used for the optimal model. We have amended these details to the Methods section of the paper.

Regarding the Reviewer's rebuttal question 4, we did not carry out parameter optimisation for the XGBoost and LightGBM comparisons. We kept all parameters as those default in each algorithm's documentation. Thus the output AUCs are from non-optimised models. The detection by both models of similar predictive TFBS combinations suggests that even the unoptimised models have similar predictive power.

3. For question 5, the authors did not respond to my concerns regarding the heterogeneity difference between cell line data and patient data. Also, why single-cell data was not considered.

It is true that there will be more heterogeneity between patients, which will harbour unique assortments of secondary mutations. Moreover, subpopulations of cancer cells within each patient may have differing mutations. Despite this, we still detect a high level of concordance between expression and long-range interaction patterns in the cell line and patient samples. Additionally, the new PML-RARA CUT&Tag data that we present in this revised version indicates that most of the PML-RARA binding sites detected in the patient are also present in the induced cell line.

While single-cell analyses are attractive to characterise subpopulation cell groups, it is incompatible with high-resolution Hi-C assays.

Reviewer #5 - Expert in CUT&RUN, CUT&Tag, and ATAC-seq (Remarks to the Author)

This revised manuscript has already been reviewed by four referees, each with different areas of expertise. I am largely confining my review to the evaluation of genomic profiling data (mainly CUT&RUN and ATAC-seq, along with CUT&Tag data provided in the rebuttal).

The CUT&RUN data appear to be of high quality. Although replicates have different numbers of peaks in some conditions, they overlap reasonably well (Fig. S2B), suggesting the difference in peak numbers may reflect a thresholding artifact that includes some low-level peaks in one replicate that are missed in another. The authors indicate that they use peaks that overlap in both replicates for downstream comparisons/analyses, which is appropriate. In the rebuttal, the authors included CUT&Tag data from NB4 cells that anecdotally appear to overlap with their U937-PR9 induced CUT&RUN data at four loci. However, to better address Reviewer 1's concern about these data, the authors should compare the overlap of their CUT&RUN and CUT&Tag peaks throughout the genome, rather than comparison of the CUT&Tag data with previous ChIP data, and it is worth including the CUT&Tag data in the supplement even if just for QC. Nonetheless, I agree with the authors that the increased signal to noise of CUT&RUN likely accounts for the 'extra' peaks relative to ChIP-seq.

We now show a genome-wide comparison of PML-RARA peaks in induced U937-PR9 cells and NB4 cells, detected by CUT&RUN and CUT&Tag, respectively. It shows that 67% of the peaks detected in NB4 cells are

also present in U937-PR9 cells (Rebuttal Figure 2A, Figure S2L). Additionally, we now provide new CUT&Tag data for PML-RARA binding in a primary APL patient sample, which shows a significant overlap with induced U937-PR9 cells of 38% ($\chi^2 = 9.7^{e-22}$) - Rebuttal Figure 2B. It demonstrates that at the very least, the bulk of binding sites that we detect in U937-PR9 cells is relevant to disease.

In addition, I agree with the authors that the lack of fixation in the CUT&RUN protocol is a positive, allowing fixation/shearing mediated artifacts to be eliminated. In addition, crosslinking has been shown to have minimal effects on CUT&RUN. Finally, the lack of peaks in the uninduced sample suggests the effect of 'leaky' PML-RARA expression is negligible. However, if this concern remains, it should be trivial to measure uninduced and induced protein levels (not mRNA) of PML-RARA by Western Blotting to put this issue to rest.

We have now carried out the Western blot, as suggested by this and other reviewers. We assess relative PML-RARA protein amounts in the uninduced and induced U937-PR9 cell line, NB4 cell line, MV4-11 cell line (which do not harbour a PML-RARA rearrangement) and three primary samples from APL patients (Rebuttal Figure 1). Our blot shows that while the induced U937-PR9 cells have high amounts of PML-RARA, the levels are closer to that which is detected in patient samples than in the NB4 cell line. This suggests that the U937-PR9 cell line is a more realistic model than NB4 cells. We highlight again that expression levels in the patient samples are variable meaning that there is no standardised protein amount among patients. It shows that the U937-PR cell line is a suitable model for PML-RARA expression for at least some patients. We note that while there is no detectable PML-RARA protein in MV4-11 cells, there is low-level detection in uninduced U937-PR9 cells. However, as evidenced by the lack of PML-RARA binding and fusion transcript in the uninduced U937-PR9 samples, measured by CUT&RUN and RNA-seq respectively, its presence has negligible effect.

The discussion of the ATAC-seq results seems to deemphasize an important finding. The authors write "While not entirely correlated, genes with PML-RARA bound promoters and differential ATAC-seq peaks were more likely to be downregulated than upregulated (Figure S4F)." However, the largest group of genes with differential ATAC peaks (essentially all of which show reduction in ATAC signal) are genes that do not change in expression. Although chromatin accessibility is thought to be one of the best markers of regulatory element activity, here, reduction of accessibility has no effect on gene expression in a large majority of cases. This may suggest that while accessibility of a regulatory element at 'steady state' (in unperturbed cells) may be a marker of gene activity, reduction of accessibility during a perturbation (induction of PML-RARA in this case) does not necessarily reveal a reduction in gene activity. The authors may want to discuss this point further.

The reviewer makes an important point. A significant population of elements whose ATAC-seq peaks are diminished do not significantly change expression. However, we observe that closing peaks do not entirely extinguish. It is possible that in a subpopulation of the cells, the peaks remain fully open. This could be addressed by single-cell ATAC-seq, although we consider this to be beyond the scope of the present study. An equally feasible possibility is that the diminishment of ATAC peak represents a change to the transcription factors that are bound to the element. The placement and configuration of the incoming TFs at these sites may provide greater protection from nuclease digestion to the chromatin than those they are replacing. We have added this commentary to the discussion.

Finally, although I am not an expert in APL, I do note there is a benefit to a study that integrates expression, binding, accessibility and 3D folding data to draw conclusions from the integrated dataset, even if one or more of each of these approaches has been previously performed in PML-RARA models. Furthermore, although it is anecdotally understood that many TF binding sites are ineffectual for activating or repressing transcription, there are to my knowledge relatively few studies where this is addressed systematically, which is also useful to the genomics and gene expression communities.

We appreciate the Reviewer's assessment of the importance of this observation and agree that it is an understudied phenomenon.

Two minor comments:

1. There are numerous figure panels in the main figures and supplement in which the text labels (including but not limited to the labels of axes) are so small they are nearly unreadable. Some examples include Figs. 1c, 3i, 3j, 4b, 5f, S2b, and S5a.

We have improved the readability of labels in the figures.

2. Line 113: "We applied the Cut&Run method to identify PML-RARA binding sites due to its low signal-to-noise ratio and..." I think you mean high signal-to noise (relative to ChIP-seq).

Yes, this is a mistake, which we have corrected.

Rebuttal Figure 1

Rebuttal Figure 1: Western blot analysis of PML-RARA binding. A) Western blot showing the PML-RARA and GAPDH (loading control) protein expression levels in whole cell lysates from NB4, three APL patients, U937-PR9 induced, U937-PR9 uninduced and MV4-11 cells. The bar plot above the blots shows the semi-quantitative expression levels calculated from each band's normalised intensities vs loading control. Each normalised quantification is further normalised relative to the U937-PR9 induced expression level, with 1 representing a 100% expression level of the induced U937-PR9 band. B) The full Western blot lanes probed for PML-RARA, the red box shows the band representing PML-RARA, just under 130kDa. Other bands within the blot represent nonspecific binding of the antibody.

Rebuttal Figure 2: Comparisons of Cut&Tag/Run for PML-RARA in U937-PR9, NB4 and an APL patient. A) Overlap of the 20,074 NB4 peaks (derived from two replicates) and the 15,412 induced U937-PR9 peaks. B) Overlap of the U937-PR9 induced and APL patient peaks. C) Overlap of the NB4 and APL patient peaks. For all Venn diagrams, χ^2 (Chi-squared test) was used to determine if the overlap was significantly greater or less than expected by chance. The expected by chance was calculated by partitioning the whole genome in 10kb bins and randomly subsampling these bins to the same size as the overlapping sets. D) Bar plot showing the fraction of aligned reads that overlapped with significantly called peaks in each dataset. E) Two Seqmonk

tracks showing the raw read build-up at two example genomic locations in the patient CUT&Tag and the induced U937-PR9 CUT&Run. The highlighted regions between the two grey lines show read build-up in the patient sample which were not called peaks, but IDR analysis shows they are consistent with stronger peaks called in the induced U937-PR9 dataset. F) Two scatter plots showing the outcome of the IDR analysis. In brief: IDR calls peaks with a low macs2 threshold ($q \leq 0.1$) in each dataset, matches common called peaks, ranks common peaks in each dataset, and plots the ranks in a scatterplot. Those peaks that have consistent ranks are considered consistent peaks. The first scatter plot shows the log₁₀ ranked scores of peaks in the patient (x-axis) vs each peak in the U937-PR9 dataset; the black dots represent the 8,119 peaks considered to be consistent based on IDR thresholding. The number of consistent peaks identified is then compared to what a 'perfect' replicate experiment might produce, this is where both samples are merged, randomly split into two 'pseudoreplicates' and the IDR plot repeated (second scatter plot). This identified 21,873 consistent peaks, which is 2.6x more than the original replicate.

Rebuttal Figure 3: Comparisons of Cut&Tag/Run for PML-RARA in U937-PR9, NB4 and an APL patient. A) Box plot showing the log₂ SEACR peak calling score for consistent peaks between the U937 and APL patient (pink) and non-consistent peaks (blue). * indicates the t-test p-value for the differences in peak scores. B) Overlap of the U937-PR9 induced and APL patient peaks, showing that the 5,829 consistent peaks overlap 2,738 gene promoters. C) Seqmonk track examples of normalised (by total read count) peaks for NB4 (blue), APL patient (red) and U937-PR9 induced (green) at two key myeloid genes. *SP1* is an upregulated example and *CEBPA* is a downregulated example. D) Barplot showing the proportion of the 2,738 consistently bound genes that are significantly upregulated (green) downregulated (red) or have no change in expression (grey) in the U937-PR9 model. E) Pie chart showing the genomic distributions of the 5,829 consistent PML-RARA binding sites.

Reviewers' comments:

Reviewer #1 (Remarks to the Author):

NOTE FROM THE EDITOR: This reviewer only provided confidential remarks to the editor, and expressed that their concerns were not addressed.

Reviewer #2 (Remarks to the Author):

I have no further comments to add beyond those already made.

Reviewer #4 (Remarks to the Author):

The authors addressed all my previous concerns.

Reviewer #5 (Remarks to the Author):

The authors have addressed my concerns.

In the following document, we will describe our perspective of the disagreements with two of the five reviewers of our manuscript for *Nature Communications*. The remaining three reviewers became satisfied through the review process. The dissatisfied reviewers are reportedly experts in APL as a disease. We value their feedback concerning the use of the model system we employed and the applicability to disease. We incorporated their recommendations on controls into the manuscript and assert that it wholly validates our approach of using the U937-PR9 inducible cell line system as a proxy for studying PML-RARA recruitment. While not acknowledged by these reviewers, we also corroborated these studies in APL patient-derived samples. Finally, we worked to better relate our findings to disease relevance.

We appreciate that certain types of experiments such as machine learning strategies can be challenging for many reviewers who are unaccustomed to their deployment in genetics research manuscripts. In these cases, it's reasonable to expect reviewers to acknowledge the limits of their expertise and defer to others who are better equipped to assess these subjects. The machine learning and downstream analysis is absolutely central to the main conclusions we make in the study. We are concerned by a complete lack of engagement with both the machine learning and patient sample analyses presented in these sections of the manuscript, in conjunction with claims that the study lacks novelty.

At the bottom of this document, we address point-by-point how we have responded to the major criticisms of the two reviewers. Where the comments have extended across review rounds, we have endeavoured to place the full response following the first point, then refer back to it at subsequent rounds.

First, however, we can provide a shortened version of what were the criticisms and our responses and actions.

Summary of criticisms and our actions to address these:

The criticisms of both reviewers centre on 1) the validity of using a cell line that reflects physiological PML-RARA binding patterns, 2) the relevance to APL and 3) study novelty.

1) Regarding the first point, we provide analysis by RNA-seq, CUT&RUN and Western blot that show PML-RARA binding patterns remain consistent across the highest expressing (induced U937-PR9), lowest expressing (NB4 cells) and intermediate expressing (ex vivo APL patient) samples. It demonstrates that the increase in detected binding sites is not due to the 35% higher protein levels compared to the patients but is a result of the improved sensitivity of the assay compared to earlier ChIP-seq experiments, which is consistent with similar experiences across the field. The interpretations from these analyses are not ambiguous and should alleviate this concern.

2) Concerning the second point as to whether the cell line can act as a suitable model for APL, we did corroborate the expression and long-range interaction patterns in ex vivo samples from two APL patients (shown in Figures 6 and Supplementary Figure 6). Neither reviewer provides any comment concerning the patient sample analysis. These experiments show a high concordance of expression between the cell line and the patient samples. Moreover, they show that most long-range interactions are conserved between the two experimental systems, validating the utility of employing the cell line model. We contend we do not oversell the parallels between the cell line model and the patients; indeed, we show examples of genes where the patterns in one or both patients do not correspond to the cell line (Figure S6H-L). In the discussion, we elaborate on this point and advise an assessment on a gene-by-gene basis. But in truth, these represent a minority of cases.

3) With the final point, we believe that the reviewers base their assertion of a lack of novelty without engaging with the paper's second half. Our study required that we carry out the assays that measure transcription, long-range interactions and PML-RARA binding to have a completely controlled and normalised experimental system. We do not hide that others have done this previously. How we use these data is novel

and significant. We demonstrate that machine-led analysis of the collection of binding sites within a regulatory element can predict how it will react functionally. Such pattern recognition is not possible using standard enrichment analysis due to the sheer complexity of binding sites within regulatory elements. Neither reviewer properly engages with this, yet it is central to the study and is a significant advance in interpreting gene regulation.

We address these three criticisms by including additional analysis and interpretation in the manuscript.

1) The previous submission included Western blot analysis and NB4 CUT&Tag experiments. We now add clarification on these experiments and their interpretation.

2) To better relate our findings to APL as a disease, we highlight in the discussion several coagulation-related genes involved in coagulation that we identified in the cell line model (and corroborated in the patient samples) as upregulated PML-RARA binding targets. The flagging of these genes is significant because bleeding events remain the greatest risk of death in APL patients through undetermined mechanisms. Our observations imply a direct role of the PML-RARA protein and have important implications for the disease.

3) In correspondence with the editor, she mentioned her reservations about developing and reporting machine learning (ML) methodologies using a single cell line. Taking this feedback on board, we agree it is necessary to demonstrate that the method is robust and applicable across other cell types. We now include the outcomes of modelling across two ex vivo cell populations, comparing normal CD34+ haematopoietic progenitor cells, from healthy individuals, with leukaemic blast cells, from APL patients (shown in a new Figure 7). We show that ex vivo cell ML modelling performs nearly as well as cell line modelling at predicting the functional activities of the regulatory elements. It is noteworthy, considering their higher cellular divergence, compared to the cell line. As before, it reveals tight clusters of similar regulatory elements with highly interconnected transcription factor networks predicted to drive the specific activities of these regulatory elements, including factors that are well-known drivers in CD34+ cells and APL. Moreover, it shows that the genes associated with these regulatory elements within these clusters significantly enrich for gene ontology biological processes central to CD34+ (e.g., stem cell maintenance) and APL cells (e.g., neutrophils, antigen presentation). It demonstrates that we can uncover new gene regulatory information on biologically meaningful sets of genes for both cell types within the comparison. In all, we believe that the inclusion of this additional ML-led modelling strengthens our conclusions and underscores the strategy's applicability.

Beyond these additions, we have prepared a supplementary table that lists the data used in the analyses of the multi-omics libraries and the ML-led modelling, which increases the data accessibility for the readers.

Point-by point responses to Reviewers for the first two review rounds

Reviewer #1 (Round 1)

Villiers et al. used integrated multi-omics and reported context-specific gene regulatory activities of PML-RARA in APL. They analyzed RNA-seq, cut&run, ATAC-seq, and Hi-C datasets, mostly in an inducible cell line model (U937-PR9). This study showed some technical advances. However, this work showed little conceptual advance, and most conclusions were similar to some work previously published. The paper is predominantly a genomics paper and lacks molecular and biological studies to confirm the findings.

While it is true that this is primarily a genomics/gene regulation paper, it demonstrably goes beyond published works. While we generated similar libraries that have been analysed by others in published work, we go on to use these and other datasets to interrogate the functional consequences of PML-RARA recruitment. We use machine learning to help build models that predict how the element behaves after recruitment of PML-RARA, based on complex transcription factor binding site (TFBS) signatures that modulate PML-RARA activity. This study is supported by molecular studies (measurements of transcription, long-range interactions and chromatin occupancy) and biological studies (corroboration of transcription and long-range interaction patterns in APL patient samples. The reviewer does not comment on these experiments or analyses.

1. This study was mainly performed on the U937-PR9 cell model, which expresses huge amount of PML-RARA and is difficult to finely tune the level of PML-RARA expression. This cell model has been associated to a number of artefacts in past studies and could clearly lead to binding to low affinity DNA sites that are not recognized in real APL cells.

In response to this point, we initially carried out an analysis of the mRNA-seq data collected in the study, quantitating the amount of fusion gene transcript in the U937-PR9 cell line, patient samples and another PML-RARA fusion-containing cell line, NB4. It showed that 1) the fusion transcript levels are variable across patient samples, and 2) similar amounts of the transcript are detected in both cell lines.

Neither Reviewer #1 nor 2 was satisfied with this analysis and therefore in the subsequent round of revision, we provided Western blot quantitation of the protein amounts of PML-RARA in the U937-PR9 cells, NB4 cells and patient samples. In these experiments, which we added to the manuscript, we showed that protein levels are also variable across patient samples. We also showed that protein levels were somewhat higher in the U937-PR9 cells, although it does not appear to be to an extreme level (ranging from 250% higher than two patients, and 35% higher than another patient). Quantitation in the NB4 cell line showed that its protein amount was considerably less than the patients (ranging from 250% for two patients and 500% higher for the other patient).

While we agree that the semi-quantitative analysis of the Western blot indicates that PML-RARA protein is more abundant in the U937-PR9 cell line, the important question is whether these differences are significant enough to alter where the protein binds. We had already presented Cut&Run analysis of PML-RARA binding in the U937-PR9 cell line. We next provided Cut&Tag analysis (an analogous method to Cut&Run) for the NB4 cell line, which expresses the protein at a lower level than both the U937-PR9 cells and the patient samples (described above). Our analyses demonstrate that 67% of the PML-RARA binding sites detected in the U937-PR9 cells are shared with NB4 cells. This represents a highly significant overlap between these two distinct cell lines ($\chi^2 < 2.2 \times 10^{-100}$). Given that their PML-RARA protein levels differ by more than 650%, with patient levels sitting in between, it appears highly unlikely that the additional PML-RARA binding sites we detect in U937-PR9 cells by Cut&Run are a consequence of excessive PML-RARA protein.

One final experiment that we conducted for the reviewers is a PML-RARA-binding Cut&Tag analysis in a single APL patient sample. We only included the analysis in the rebuttal for the reviewers because we were only able

to carry out one non-replicated experiment and as such, it lacked the statistical rigour required for publication. This library had a higher level of non-specific background than was obtained with the other libraries, likely due to the fragility of patient samples in this assay type, resulting in fewer statistically significant peaks. However, we detected a strong overlap of PML-RARA binding sites compared to both U937-PR9 cells (38%) and NB4 cells (36%). This is a high overlap considering it's based upon a single replicate. Irreproducibility Discovery Rate analysis indicated that many peaks that overlap with the cell lines only just miss the statistical threshold and would likely be visible if we could have included a replicate library in the analysis.

Taken together, the further analyses that we provide the reviewers indicate that the PML-RARA protein levels present in the U937-PR9 cells are not hugely different to what is detected in patient samples. Moreover, the amount of protein does not appear to affect the binding profile, seeing as low-level expression systems show largely the same binding patterns as high-level expression systems.

2. The finding of context-specific gene regulatory activities of PML-RARA in APL is not novel. Tan et al. have reported the dual function of PML-RARA in APL using the multi-omics approach in NB4 cells (PMID: 32854112). They have knocked down the PML-RARA in NB4 cells and identified the dual function of PML-RARA in transcriptional regulation in APL, which is similar to the conclusion in this study.

We pointed out to the reviewer that we are aware that like us, Tan et al identified APL binding targets that are upregulated and we cite their work. However, they did not note the vast majority of APL targets (80%) that do not respond transcriptionally to the binding. More importantly, this observation is not the conclusion of the study. We use a machine learning strategy to identify transcription factor binding site (TFBS) signatures that can predict how a regulatory element will behave upon PML-RARA recruitment. This is the major feature of the manuscript, yet the reviewer provided no comment.

3. The concept of the engagement of PML-RARA in chromatin conformation regulation has also been reported recently by Wang et al (PMID: 32393309). They have also applied the U937-PR9 model and investigated the impact of PML-RARA on chromatin interaction using the CHIA-PET approach. They reported the repression function of PML-RARA in chromatin conformation regulation.

This point is similar to the previous one. We are aware that Wang et al also detect long-range interactions with PML-RARA binding and we cite this work and do not claim that ours is an original observation. However, we go on to use our long-range interaction data as a key behavioural feature of PML-RARA-binding elements, as we identify elements that either gain or lose interactions upon PML-RARA recruitment. This forms a central classification that is crucial to developing machine-learned models of gene regulatory TFBS signatures, which is not commented upon by the reviewer.

4. Another major issue is that there are a number of published studies which are directly relevant to this manuscript, but the authors do not properly give credit to these other authors for their contributions.

We queried this point as we did not knowingly omit any key references. We asked the reviewer to provide examples, yet none were given, and the reviewer did not follow up on this criticism.

5. Most results were descriptive data analysis. The author should provide evidence on what factors determines the transcriptional repression or transcriptional activation activity of PML-RARA; what determines the enhanced or decreased chromatin interaction during PML-RARA induction; whether is

related to the post-translational modification of PML-RARA; and what is the interaction with well-known co-factors (such as RXR)?

We pointed out to the reviewer that while post-translational modification of PML-RARA may also contribute to the differential transcriptional output of PML-RARA bound genes, our study has focussed on the cooperating binding sites that modulate PML-RARA activity. The identification of the transcription factors that bind to these sites represents a major undertaking that extends well beyond the scope of this manuscript and would significantly increase its size beyond that of a standard submission. Moreover, it would detract from the central message of the paper that demonstrates the application of machine learning to distil complex transcription factor binding site patterns. The main message of the paper is that TFBS compositions that determine the behaviours of PML-RARA binding elements are complex and there are multiple conducive environments that contribute to a defined behaviour.

Reviewer 1 (Round 2)

The authors have not really addressed my concerns about the conceptual advances of their findings over earlier works and the functional or experimental evidence of their conclusions that can be drawn at this stage.

The reviewer seems not to accept that we present anything novel beyond the works of Tan et al. (gene activation by PML-RARA recruitment) and Wang et al. (long-range interactions in the context of PML-RARA binding). There is essentially no engagement with any analyses that we carry out downstream of data collection in the cell line. Beyond the genes that are either up or downregulated, we identify a third class that does not respond transcriptionally to PML-RARA binding. We show that these three classes can be segregated further based on their gains or losses of long-range contacts. We use these classifications to train and test machine-led models that can successfully predict using the complex TFBS signatures how a regulatory element will behave upon PML-RARA recruitment. We show that the TFBS signatures display a high representation of binding sites for transcription factors (TF) that are present in physical interaction networks, including TFs that are highly relevant to APL. Genes whose elements are similar by composition display gene activity functional enrichments, which is suggestive of a coordinated transcriptional response. None of these experiments is discussed or acknowledged by the reviewer in any way.

The reviewer also does not appear to consider the analyses we carry out, which show that the vast majority of expression and long-range interaction patterns that we detect in the U937-PR9 cell line system are consistent with samples derived from APL patients.

Particularly, the expression level of PML-RARa (Rebuttal Figure 1A) seems incorrect in U937-PR9 cells including both before and after induction.

Here the reviewer refers to the fusion transcript quantitation from the RNA-seq data. We reverified these analyses and can confirm they are accurate. The data are available for corroboration.

The authors ignored the assays on the protein levels.

As described above in the response to the Reviewer's first point from Round 1, we did provide protein level analysis by Western blot that shows a PML-RARA is not hugely over-expressed in U937-PR9 cells.

The validation of PML-RARa binding sites should be compared between PR9 and NB4 cells (or patient samples).

Also described in the same section, we provide a binding site comparison with NB4 cells, which shows a high overlap between the two cell types. We also carried out a preliminary experiment (again, as described above) in a patient sample, which shows the binding patterns are consistent with the cell lines.

Current validation between chip-seq and CUT-and-Tag in NB4 cells does not support the authenticity of PML- RARa binding sites.

We have provided ChIP-seq and Cut&Tag/Cut&Run analysis in NB4 and U937-PR9 cells. In each case, we detect vastly more sites in the Cut&Tag/Run experiments. This is likely due to the widely accepted view that the new methods have a considerably lower background than ChIP. Also, we used an antibody specific to the PML-RARA fusion protein, in contrast to ChIP-seq experiments by others, where overlapping binding profiles of PML and RARA were used to identify PML-RARA binding sites.

Also, the author did not provide the analyzed data with Supplemental Tables as well as the raw data (only RNA-seq data under the GEO accession GSE173754, the token for cut and run data is not available).

We rectified this error.

More importantly, authors claimed the sensitivity of Cut-and-Run which helped them find that the fusion protein was recruited to sites where other factors were already bound, but they denied the importance of identifying the factors which influence the PML-RARa binding and determine the transcriptional outcome.

We point out that the identification of the transcription factor binding environment at these PML-RARA recruitment sites that may modulate its activity is precisely the point of the second half of the paper. We focus on the transcription factor binding sites rather than the putative transcription factors themselves. We point out that multiple TFs are capable of binding to a given binding site and speculatively chasing specific factors that might be recruited would quickly descend into a time-consuming and expensive endeavour. Our machine-learning-led characterisations of the PML-RARA binding sites show that the binding environment composition and syntax are highly complex. It simply is not within the scope of this study to carry out tens of ChIP-seq/CUT&RUN assays to try to assemble the factors involved at these sites.

Reviewer #2 (Round 1)

In this report, the authors have explored an inducible model of PML/RARA expression to explore, in a very extensive manner, the genomic changes resulting for expression of this oncoprotein. This involves RNA-seq, ATAC-seq, cut and run and capture Hi-C. As agreed with the editor, this review will not concentrate on the genomics analyses by themselves, but on the importance of these studies for our understanding of the APL model and pathogenesis.

First, the cellular model, while technically convenient, has a number of serious caveats. It is an immortalized cell-line, the promoter leaks so that PML/RARA expression can be detected at low levels before Zn administration and the level of expression post-Zn induction is much higher than in APL cells, precluding physiological relevance. The system is also polluted by the very strong Zn-initiated stress response clearly visible on Fig. 1B.

Part of this first comment regarding levels of PML-RARA expression after zinc induction relates to comments raised by Reviewer #1. Please refer to the detailed response (found in Q1 of Reviewer #1, Round 1) that describes the Western blot and Cut&Run/Tag experiments in U937-PR9, NB4 and patient samples. It shows

that PML-RARA protein levels are not greatly higher than that found in patient samples. Moreover, the binding patterns remain consistent across all samples.

Regarding the supposed leakiness of the U937-PR9 cells, this idea comes from the examination of Western blot data, where a faint and diffuse band is detectable in uninduced cells. This observation is not corroborated by our RNA-seq data where no PML-RARA fusion transcripts are detected in uninduced U937-PR9 cells, in contrast to induced cells, NB4 and patient samples. Nor do we detect significant PML-RARA genomic binding in uninduced U937-PR9 cells by Cut&Run. We do see the faint and diffuse band by Western blot in the uninduced sample, as described by others. To our eyes, the bands do not exactly match the PML-RARA bands in other lanes, which may be indicative of non-specific bands, although this is open to interpretation. Western blots are semi-quantitative. The presence of PML-RARA in uninduced cell samples would be incongruent with both our RNA-seq data, where no fusion transcripts were detectable and our Cut&Run data, which demonstrated virtually no PML-RARA peaks. It is conceivable that little PML-RARA protein is present, and its binding is negligible.

We indeed observe a strong upregulation of genes that are responsive to zinc, which we acknowledge in the manuscript. However, we disagree that this amounts to 'pollution' that would skew the results across the genome. Indeed, functional enrichment analysis shown in Figure 1 of the manuscript demonstrates that the genes that are most impacted transcriptionally are those involved in myeloid differentiation and cancer. We corroborated our RNA-seq data with patient samples and showed a high degree of concordance for differentially expressed genes.

There is a very strong correlation between the induced cell line and patient-derived samples for transcription (measured by RNA-seq) and long-range promoter interactions (measured by promoter capture Hi-C). Moreover, in response to the reviewers' comments, we provide Cut&Tag PML-RARA binding analysis in NB4 cells (included in the manuscript) and an APL patient sample (for the reviewers' assessment, as discussed in the Reviewer #1 section), which demonstrates a strong overlap of binding profiles. Collectively, it shows that expression, long-range interactions and PML-RARA recruitment are not reliant on exposure to zinc.

Second, multiple previous studies have reported on this issue, and it is unclear what is conceptually new in this report, except that more technologies were used. At large, all published genomics studies have demonstrated that PML/RARA is an altered transcription factor but failed to identify critical determinants of pathogenesis. The Wang study, in Blood, already reported that PML/RARA can be an activator of a significant number of genes. In that sense, the abstract is somehow misleading when stating that this issue is "under-explored". It is actually unclear to this reviewer what are the novel conclusions drawn from this study?

Like Reviewer #1, he/she appears focused on the similarities of our study to others, rather than examining what is novel and notable. It is noteworthy to us that in his/her study description (at the top of the response) there is a disclosure of an intended lack of engagement with the genomics analyses. This is evident because there is no consideration of the downstream analysis where we employ machine learning on integrated multi-omics datasets to identify distinct TFBS signatures that direct different functional behaviours of the PML-RARA binding regulatory elements. Our study does indeed indicate how different TFBS signatures can modulate PML-RARA activity across at least six distinct functional behaviours.

Surprisingly there was also no comment on the transcriptional and long-range interaction analyses we carried out in APL patient samples, which show that the majority of expression and regulatory contacts that we detect in the cell line model are also present in APL patient samples. Given that the reviewer showed concerns about the physiological relevance of the cell line model to APL, we feel this is problematic.

Third, it is unclear to what extent many of the loose correlations or overlaps shown between gene sets or binding sites, only reflect similar differentiation status.

The distinct ontological enrichments we identify are not based on loose correlations, but on stringent, statistical significance. If there is a similar differentiation response amongst these genes, it is likely driven by these shared regulatory inputs.

Reviewer #2 (Round 2)

I am unconvinced by many of the points put forward in response to the comments of reviewers 1 and 2. The U937 system has been around for almost 30 years and many groups have used it. All reported much higher levels of PML/RARA expression by Western blot, when compared to APL patients or NB4. The mRNA assessment by the authors is likely not reflecting protein levels.

As described above for Reviewer #1, we have included Western blot analysis, which shows that in the conditions in which we have employed the assay (five-hour induction $100\mu\text{m Zn}_2\text{So}_4$), we do not detect a much higher level of PML/RARA expression when compared to patient samples. Moreover, NB4 cells and an APL patient sample that have lower levels of PML-RARA protein display highly similar PML-RARA binding profiles to the induced U937-PR9 cells. Therefore, alterations in protein concentration do not impact PML-RARA recruitment behaviour.

Similar to reviewer 1, my major comment remains: what do we actually learn from these studies? It is textbook knowledge, for example for nuclear receptors, that transcription binding does not necessarily translate into transcriptional control. That an altered transcription factor, with a high affinity for DNA (because of the PML-mediated dimerization of the RARA/RXRA complex) and reduced binding site specificity engages into different types of chromatin interactions is hardly surprising. The dual effect (activation or repression) of PML/RARA on some of its targets was the main message of a very detailed previous publication in Blood.

We do not agree that this transcription factor binding without a transcriptional effect is textbook knowledge and are unaware of any reports where this has been characterised in any detail. Of note, Reviewer #5, who was brought into the review process to interrogate the Cut&Run and Cut&Tag experiments, touched on the importance of this observation, commenting that while perhaps anecdotally observed, it has not previously been studied systematically, using complementary measures of transcription, as we present in this manuscript.

Again, the reviewer is questioning what is learnt from these studies without any engagement with the second half of the paper. We demonstrate that PML-RARA recruitment to regulatory elements can display a remarkable diversity of transcriptional and long-range interaction behaviours. We use a novel application of a machine learning method to deconvolute complex TFBS environments to identify signatures that can predict how the element will behave upon PML-RARA recruitment. It provides novel insight into distinct programmes in which PML-RARA can operate. We show that the majority of these elements are engaged in the same interactions in APL patient samples. All of this is novel yet bypassed by the reviewer.

It is not because the authors acknowledge the overwhelming Zn response, that this not does blur the data.

We provide our response to this comment in the first round of Reviewer #2 comments (please see above).

local transcription factor binding site environment at PML-RARA bound positions to reveal distinct signatures that modulate how PML-RARA directs the transcriptional response (abstract)...

What is the actual evidence for that? The machine learning signatures? The authors describe them as very complex, suggesting that they do not have much biological relevance? It is the only novelty put forwards in the abstract.

The evidence for this is that the model correctly predicts the behaviours of the different elements based on their transcription factor binding site compositions. The signatures are distinct between the functional classes and between subsets within classes. Yes, the number of features used by the machine learning algorithm to correctly predict the functional outcome of the element is large, implying that the signature is complex. However, there is no suggestion that the signature has little biological relevance. Indeed, the fact that the prediction is dependent upon so many modulating elements that contribute incrementally is in itself a significant observation. It is also notable that the signatures often corresponded to networks of TFs with known interactions, as evidenced by the STRING analyses. They include well known PML-RARA binding partners such as SP1 and MYB, both which have been implicated in APL.

Comments of APL clinical aspects are not accurate. The key remaining issue in clinical trials are the (rare) early deaths.

It's not been clear to us which aspects we've discussed are inaccurate and we acknowledge that early death is the major issue. Specifically, the greatest risk relates to coagulopathy and haemorrhagic events. In the final round of revisions, we included discussions on a group of anti-coagulation genes that emerged from our U937-PR9 analyses, which recruit PML-RARA in conjunction with gained long-range interactions and increased expression. Both the expression and long-range interaction profiles for these genes are consistent in the APL patient samples. It provides new indications that these genes, and by extension coagulopathy may come under direct control by PML-RARA, which may have important clinical implications.

The authors do not discuss the possibility that many of the long-range interactions are a mere reflection of the changing differentiation profile.

Firstly, uninduced U937-PR9 cells are not differentiating. Secondly, it is well established that the expression of PML-RARA creates a differentiation block, so it's highly unlikely that it will alter the non-differentiating status upon induction. Thirdly, the cells are induced for only five hours before assaying. In such a short period, none of the cells will have undergone a full cell cycle.

Additionally, any alterations to long-range interactions that occur through differentiation reflect the rewiring of promoter contacts to distal regulatory elements that shape the transcriptional response, rather than a generic side effect.

REVIEWERS' COMMENTS

Reviewer #4 (Remarks to the Author):

In this revision, the authors defended some criticisms raised by other reviewers and added a new experiment to demonstrate the robustness and application of their method in other samples. Since I am not an expert in AML, I only comment on the computation and data interpretation parts. Overall, I stand with the authors regarding the validity of using cell line data and computational novelty. I also raised similar concerns in the first two rounds and was satisfied with the authors' responses and edits. The authors have proved that the cell line data share parallel information with patient samples, and they have acknowledged the potential pitfalls, which are harmless to their model training. Although the authors did not develop the machine-learning tools, the way they used them for training is new. Regarding the new experiments, I only have several minor comments that the authors could adapt to help readers better understand the story.

- 1) Figure 7B and 7C can be reversed so that people can compare the clustering results with and without modeling training.
- 2) Add specific subcluster numbers next to clusters in 7D. Something like the figure below. I was confused when only looking at the figure and legend. Also, Figure 7D is not the same figure as 7C. Some individual dots are missing. Not sure if the reason for using different resolutions.

- 3) Should it be SHAP weighting or SHAPLEY weighting or SHAPLEY interpretation score? Better to be consistent for the figure legend and the manuscript.
- 4) I believe it is not recommended to include color-related descriptions in the figure legend. Better to add any legends in the figure.
- 5) We need some clarifications for selecting those subclusters in Figures E & F, either statistically (e.g., clusters with top XX scores) or biologically.
- 6) Including detailed descriptions of the interactions, network, TFs, and pathways would be appreciated. Highlight any shared findings in conjunction with the U937-PR9 analysis.

Reviewer #6 (Remarks to the Author): Expert in APL clinical research and genomics

In this manuscript, Villiers et al, using integrated multi-omics

datasets, characterized the influence of PML-RARA binding on gene expression and regulation in an inducible PML-RARA cell line model and primary APL patient samples.

My general feeling is that the Authors adequately addressed the comments raised by the 2 previous Reviewers, by providing sufficient evidence on the controversial topics, specifically regarding adequacy of the cell line models used for their analysis. Particularly, the Authors offered additional experiments and analyses to substantiate the statements made in response to Rev^o1 and 2 comments.

While previously speculated, PML::RARA does not uniformly lead to loss of acetylation and expression, but contextual factors may also have an impact on the transcription output, and this is one of the strengths of their in-depth analysis.

As to the machine learning approach, this paper report a ML-based method to deconvolute the several TFBS, and identify signatures predicting responses upon PML::RARA recruitment, in both cell lines and patient samples.

eXtreme Gradient Boosting (XGBoost) is a good method to perform such kind of analysis and the authors are clearly able to leverage ML-based techniques as shown in the methods and accuracy of analysis (e.g., use of an 80:20 ration for training/validation) and data visualization (e.g., tSNE, Interaction landscape plots, cluster analysis).

Additionally, just few suggestions for the authors:

1. The fusion transcript should be written according to the new nomenclature (see also PMID: 34615987):PML::RARA
2. Line 107: acute myeloid leukaemia (should be uniform to the American spelling used throughout the manuscript)
3. Line 273-274: Please adjust the grammar of this 2 sentences (the first is a fragment): “While predictive scores were lower than for the pairwise comparisons, ranging from AUC scores of 0.55 to 0.70. Our results suggest that a learned pattern could be acquired and highlight that TFBS signatures can uniquely drive a category (Figure 5C).”

Response to Reviewers

We thank both Reviewers #4 and #6 for their thoughtful comments and suggestions. We respond to each suggested alteration below.

Reviewer #6 (Remarks to the Author): Expert in APL clinical research and genomics

Additionally, just few suggestions for the authors:

1. The fusion transcript should be written according to the new nomenclature (see also PMID: 34615987):PML::RARA

We have replaced PML-RARA with PML::RARA throughout the manuscript, as suggested.

2. Line 107: acute myeloid leukaemia (should be uniform to the American spelling used throughout the manuscript)

We have amended the spelling.

3. Line 273-274: Please adjust the grammar of this 2 sentences (the first is a fragment): “While predictive scores were lower than for the pairwise comparisons, ranging from AUC scores of 0.55 to 0.70. Our results suggest that a learned pattern could be acquired and highlight that TFBS signatures can uniquely drive a category (Figure 5C).”

We have improved the grammar. These two sentences now read:

Predictive scores were lower than for the pairwise comparisons, ranging from AUC scores of 0.55 to 0.70. Our results suggest that patterns can be learned and highlight that TFBS signatures can uniquely define a category (Figure 5C).

Reviewer#4

Regarding the new experiments, I only have several minor comments that the authors could adapt to help readers better understand the story.

1) Figure 7B and 7C can be reversed so that people can compare the clustering results with and without modeling training.

Our preference is to retain the order since the main point is to compare panels 7C and 7D, since 7D is a representation of 7C, separated into defined clusters using the DBSCAN algorithm. We recognize that we have not provided information on how we identified clusters within the machine learning tSNEs of Figures 5 and 7. We now include this information in the Methods section.

2) Add specific subcluster numbers next to clusters in 7D. Something like the figure below. I was confused when only looking at the figure and legend. Also, Figure 7D is not the same figure as 7C. Some individual dots are missing. Not sure if the reason for using different resolutions.

We agree that Figure 7D (and by extension, Figure 5E) needs improvement. We intend the reader to consult the colour of the functional categories displayed in Figure 7C (and Figure 5D) when examining the individual clusters in Figure 7D (and Figure 5E). Any dot in Figure 7C (and Figure 5D) that cannot be assigned to a cluster is removed from display in Figure 7D (and Figure 5E) - this has

been reiterated in the figure legend. The resolutions in Figures 7D and 5E are different because the points are comprised of the number assignment of that cluster. We agree that this is not terribly helpful and have replaced the numbers with dots, in conjunction with a number over the cluster, as suggested by the reviewer.

3) Should it be SHAP weighting or SHAPELY weighting or SHAPELY interpretation score? Better to be consistent for the figure legend and the manuscript.

We have changed it to SHAPELY weighting, to be consistent across the manuscript.

4) I believe it is not recommended to include color-related descriptions in the figure legend. Better to add any legends in the figure.

We intended the readers to refer to the colour schemes in Figure 7B (and Figure 5C), however, we now include a colour reference to Figure 7C (and Figure 5D).

5) We need some clarifications for selecting those subclusters in Figures E & F, either statistically (e.g., clusters with top XX scores) or biologically.

We delve into specific clusters in Figures 7E&F to illustrate relevant and biologically meaningful features for these cell types. As we describe in the final paragraph of the Results section, 'stem cell maintenance' is a relevant feature in CD34+ cells. In the APL samples, we highlighted clusters with enrichments for neutrophil biology and antigen processing, both of which are defective in APL.

6) Including detailed descriptions of the interactions, network, TFs, and pathways would be appreciated. Highlight any shared findings in conjunction with the U937-PR9 analysis.

We have included in the discussion comment on how many of the most predictive TFBS are implicated in APL and appear to influence PML::RARA behavior through intricate combinations. While we did not pinpoint clusters that are shared between the APL patient vs CD34+ and the U937-PR9 analyses, this is not necessarily unexpected since the inclusion of the CD34+ comparator would naturally focus on different elements as it is a distinct cell type.